# Watermarking Autoregressive Image Generation

**Nikola Jovanović** [1,2,*]   **Ismail Labiad** [1,3]   **Tomáš Souček** [1]   **Martin Vechev** [2]   **Pierre Fernandez** [1]

## Abstract

Watermarking the outputs of generative models has emerged as a promising approach for tracking their provenance. Despite significant interest in autoregressive image generation models and their potential for misuse, no prior work has attempted to watermark their outputs at the token level. In this work, we present the first such approach by adapting language model watermarking techniques to this setting. We identify a key challenge: the lack of reverse cycle-consistency (RCC), wherein re-tokenizing generated image tokens significantly alters the token sequence, effectively erasing the watermark. To address this and to make our method robust to common image transformations and removal attacks, we introduce a custom tokenizer-detokenizer finetuning procedure that improves RCC and a watermark synchronization step. As our experiments demonstrate, our approach enables robust watermark detection with theoretically grounded p-values.

## 1 Introduction

Autoregressive models are powerful frameworks for understanding and generating diverse content types. By converting multiple modalities into discrete representations via custom tokenizers (Razavi et al., 2019; Van Den Oord et al., 2017; Zeghidour et al., 2022), a single transformer is able to seamlessly process multiple domains, including text, images (Chameleon Team, 2024; Chern et al., 2024; Tian et al., 2024a; Wu et al., 2024), audio (Borsos et al., 2022; Défossez et al., 2024), and even molecules (Hsu et al., 2022). Following patterns observed in large language models (LLMs), established scaling laws (Aghajanyan et al., 2023; Henighan et al., 2020; Shukor et al., 2025) demonstrate that the performance of these models improves predictably with size

and computational resources, leading to increasing adoption across research and industry (DeepMind, 2023a; Meta, 2025; OpenAI, 2024). Most notably, in the image domain, autoregressive models are widely studied as an alternative to diffusion models for high-quality generation (Ramesh et al., 2021; Sun et al., 2024; Tian et al., 2024b; Yu et al., 2022).

**Watermarking generative model outputs.** Regardless of the specific method, the widespread deployment of high-quality generative models has made the detection of AI-generated content increasingly challenging. This has raised significant concerns about misuse, including deepfakes, harmful content generation, and intellectual property violations. One promising direction to help address these issues is *generative AI watermarking*, in which the model provider proactively embeds imperceptible signals into generated content to verify its origin later, even under common transformations or attacks.

Recent research in this area can be categorized into post-hoc methods, which modify generated content in a model-agnostic way (Bui et al., 2023a; Chang et al., 2024; DeepMind, 2023b; Fernandez et al., 2024; San Roman et al., 2024; Zhu et al., 2018), and modality-specific generation-time methods, which alter the generation process of a specific model (Aaronson & Kirchner, 2023; Dathathri et al., 2024; Fernandez et al., 2023b; Kirchenbauer et al., 2023; Wen et al., 2023; Yang et al., 2024). The latter are the standard in LLM watermarking, offering theoretically grounded watermark detection with provably low false positive rates. However, most image watermarking research focuses on diffusion, and no prior work has attempted to adapt LLM watermarks to other token types, which could be a way to enable watermarking for autoregressive image generation. This motivates our key question:

*Can we robustly watermark autoregressive image generation models at the token level?*

**This work.** To answer this question, we adapt LLM watermarks to autoregressive image generation, watermarking image tokens together with text as illustrated in Fig. 1. We identify and address a key technical challenge. Namely, while image tokenization is designed to be *forward cycle-consistent*, i.e., tokenizing and detokenizing an image does not significantly alter it, *reverse cycle-consistency* (RCC) is often violated—we show that decoding model-generated to-

---

[*]Work done during a Meta internship. Latest version: https://arxiv.org/abs/2506.16349. Code: https://github.com/facebookresearch/wmar [1]Meta FAIR [2]ETH Zurich [3]Université Paris-Saclay. Correspondence to: Nikola Jovanović <nikola.jovanovic@inf.ethz.ch>.

*Non-archival presentation at ICML 2025 Tokenization Workshop (TokShop)*, Vancouver, Canada. 2025.

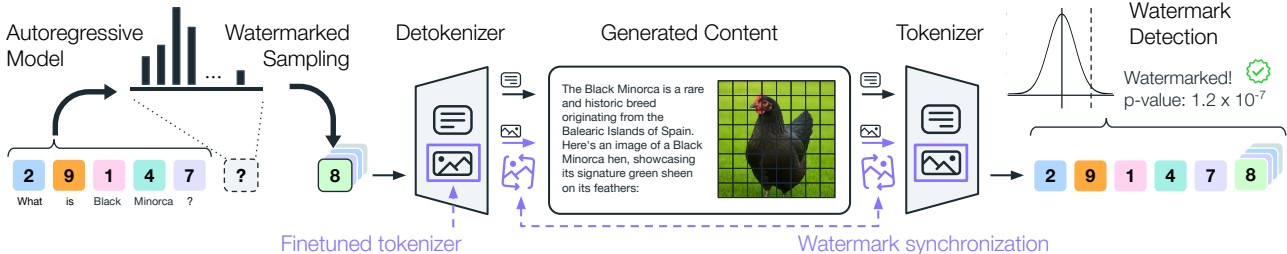

Figure 1: We watermark autoregressively generated images together with text in a theoretically-principled way by adapting LLM watermarking. We identify and address the novel challenges present in this setting (Sec. 3) via a custom (de)tokenizer finetuning procedure (Sec. 3.1) and a watermark synchronization layer (Sec. 3.2).

kens and then re-tokenizing the resulting image leads to, on average, one-third of the tokens being different. The tokens differ even more if the images are transformed between generation and watermark detection (e.g., JPEG compressed or cropped), a common scenario in practice. While RCC may not be crucial for model performance, it is important for achieving strong and robust generation-time watermarking.

To mitigate this, we adopt two main strategies, shown in Fig. 1. We introduce a lightweight finetuning procedure that optimizes the detokenizer and tokenizer to be more reverse cycle-consistent, improving watermark power and robustness to valuemetric transformations (e.g., JPEG) and attacks such as diffusion purification (Nie et al., 2022) and neural compression (Ballé et al., 2018; Cheng et al., 2020; Esser et al., 2021; Rombach et al., 2022). To then improve robustness to geometric transformations (e.g., flips), we introduce a complementary post-hoc *watermark synchronization* step, repurposing localized watermarking (San Roman et al., 2024) to detect and revert geometric transformations and recover original tokens. As our experiments show, this results in a watermark that is quality-preserving, effective, and robust. To inspire future work, in Sec. 5 we take first steps to extend our approach to audio, another modality where autoregressive generation via tokenization is popular (Borsos et al., 2022; Copet et al., 2024; Défossez et al., 2024; Nguyen et al., 2025).

**Contributions.** We make the following key contributions:

- We conduct the first study of watermarking for outputs of autoregressive image generation models, adapting LLM watermarks to obtain theoretically-principled p-values. We notably identify a key technical challenge, the lack of reverse cycle-consistency (RCC) (Sec. 3).

- We propose a finetuning procedure for image tokenization that improves RCC and significantly increases watermark power and robustness (Sec. 3.1).

- We introduce a post-hoc watermark synchronization step, which achieves geometric robustness by leveraging off-the-shelf localized watermarking (Sec. 3.2).

- In our thorough experimental evaluation across several settings, we show that our watermark is strong, quality-preserving, and persistent under a range of valuemetric and geometric transformations and attacks including diffusion purification and neural compression (Sec. 4).

## 2 Background and Related Work

**Autoregressive image models.** A long-studied approach to image generation, and the focus of our work, is to first learn an image tokenizer and then train a (conditioned) autoregressive model $\mathcal{M}$ such as a transformer, to create images by generating corresponding token sequences. Notable examples of such models include DALL-E (Ramesh et al., 2021), Parti (Yu et al., 2022), VAR (Tian et al., 2024b), and others (Sun et al., 2024; Tschannen et al., 2024; Wang et al., 2024; Yu et al., 2024). This approach is central to models for interleaved multimodal generation (Chern et al., 2024; Ge et al., 2023; Lin et al., 2024; Liu et al., 2024a; Tian et al., 2024a), such as Chameleon (Chameleon Team, 2024), AnyGPT (Zhan et al., 2024), or Janus (Wu et al., 2024). In this work, we do not consider other models that use continuous representations or combine diffusion with autoregressive mechanisms (Fan et al., 2024; Li et al., 2024; Ma et al., 2024; Xie et al., 2024; Zhou et al., 2024).

**Tokenization.** Formally, for a target modality $m$ (in this work primarily text or image, but also audio in Sec. 5), a *tokenizer* $\mathcal{T}_m$ maps each data sample $x$ to a sequence of integer tokens $s = (s_1, \ldots, s_T) \in V^T$, where $V$ is the predefined vocabulary. The *detokenizer* $\mathcal{D}_m$ attempts to reverse this process. Most text tokenizers are based on *byte-pair encoding* (BPE) (Gage, 1994). While alternative approaches have been explored (Li et al., 2024; Tian et al., 2024b), the tokenization of images overwhelmingly relies on vector quantization (VQ) (Gray, 1984; Li et al., 2024). Most models use VQ-VAE (Razavi et al., 2019; Van Den Oord et al., 2017) or its variants VQGAN (Esser et al., 2021), ImprovedVQGAN (Yu et al., 2021), and FSQ (Mentzer et al., 2023). VQ tokenizers generally consist of an encoder network $E$ and a quantizer $Q_C$. $E$ maps $x$ to a sequence

| Unwatermarked Generation | Tokens ($p = 0.73$) | Watermarked Generation | Tokens ($p = 10^{-11}$) |
|---|---|---|---|

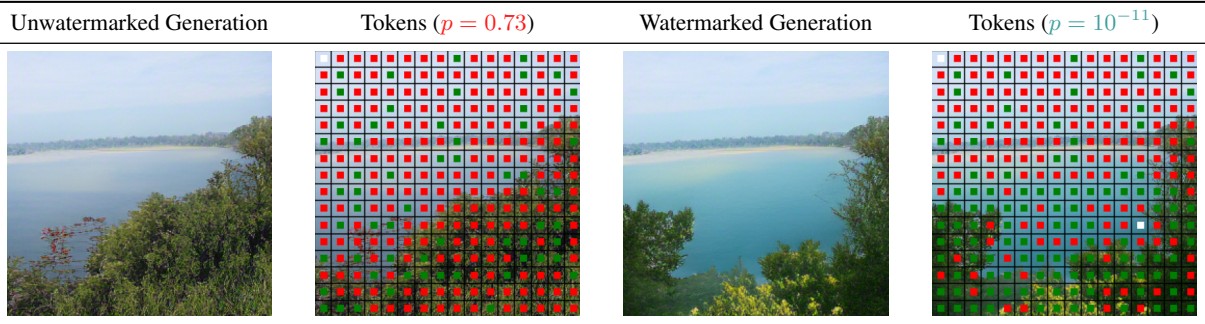

Figure 2: Example of our watermark on an autoregressively generated image. We generate the upper half of the image without the watermark. We then complete the bottom half in the same way (*left*) or with the watermark (*right*). The overlay indicates generated image tokens detected as green (■), red (■), or ignored as a duplicate (□). The watermark only alters semantics and could be detected even when applied only partially as in this case.

of *soft latents* $z = E(x) \in \mathbb{R}^{T \times d}$. Then, $Q_C$ replaces each $z_i$ with the index of the nearest entry in a *codebook* $C \in \mathbb{R}^{|V| \times d}$ to obtain discrete tokens $s \in V^T$:

$$s_i = Q_C(z_i) = \underset{j \in \{1,\dots,k\}}{\arg \min} \|z_i - C_j\|_2^2. \quad (1)$$

The detokenizer $\mathcal{D}_m$ replaces each $s_i$ with the corresponding $\hat{z}_i = C_{s_i}$ (*hard latents*), and then applies a decoder network $D$ to obtain the detokenized sample $\hat{x} = D(\hat{z})$. All components ($E$, $C$, and $D$) are typically trained jointly, primarily with the reconstruction objective.

**Watermarking AI-generated outputs.** *Generation-time methods* directly alter generations to embed statistically detectable patterns, and are the standard for LLM watermarking (Aaronson & Kirchner, 2023; Christ et al., 2023; Kirchenbauer et al., 2023; Kuditipudi et al., 2023). Similar approaches also exist for diffusion models (Fernandez et al., 2023b; Wen et al., 2023; Yang et al., 2024). In contrast, *post-hoc watermarks* modify previously generated outputs in a modular model-agnostic way, by paraphrasing text (Bahri & Wieting, 2025; Chang et al., 2024; Zhang et al., 2024a) or altering image pixels (Bui et al., 2023a;b; Jia et al., 2021; Luo et al., 2020; Ma et al., 2022; Tancik et al., 2020; Zhu et al., 2018). *Multi-bit* methods embed messages into data, sometimes in a *localized* way (San Roman et al., 2024; Sander et al., 2025), e.g., a message can be extracted from each pixel. While post-hoc watermarks have broad applicability, generation-time approaches that introduce semantic changes to the content often offer superior robustness to attacks such as diffusion purification (Saberi et al., 2024) and provide provable, key-based guarantees on false positive rates (unlike neural methods, where recovered bits may be biased or correlated (Fernandez et al., 2023b, App. B.5)). To the best of our knowledge, no prior work targets generation-time watermarking for autoregressive image models.

**LLM watermarking.** We focus on the KGW watermarking scheme (Kirchenbauer et al., 2023) (green/red water-

marking). At each step $i$ of generation, this method uses a secret key $\xi$, and previous $h$ tokens of *context* $s_{i-h:i}$ to pseudorandomly partition the vocabulary $V$ of the tokenizer into $\gamma|V|$ *green* tokens $\mathcal{G}_i$ and other *red* tokens $\mathcal{R}_i$. The logits corresponding to $\mathcal{G}_i$ are then increased by $\delta$, the *watermark strength*. The watermark detector computes the *score* $S = \sum_{i=h+1}^{T} \mathbb{1}(s_i \in \mathcal{G}_i)$ as the number of green tokens in the given sequence of $T$ tokens. Under the null hypothesis $\mathcal{H}_0$ (no watermark), $S$ follows a binomial distribution with parameters $T - h$ and $\gamma$. The p-value is calculated as:

$$\text{pval}(S, T, h, \gamma) = \mathbb{P}\left(X \geq S \mid X \sim B(T - h, \gamma)\right), \quad (2)$$

where $B(\cdot, \cdot)$ denotes the binomial distribution. A low p-value proves that the content was generated with $\mathcal{M}$. More details are provided in App. B.

## 3 Watermarking Autoregressive Image Generation

In this section, we present our approach to watermarking autoregressive image models. We identify and address the key challenge of low *reverse cycle-consistency (RCC)* via tokenization finetuning (Sec. 3.1) and watermark synchronization (Sec. 3.2). As our experiments in Sec. 4 demonstrate, this leads to a strong and robust watermark that does not affect generation quality.

**Setting.** A model provider (Alice) deploys an autoregressive model $\mathcal{M}$ that may generate arbitrarily interleaved text and images, using a tokenizer $\mathcal{T}_m$ and detokenizer $\mathcal{D}_m$ for each modality $m \in \{\text{text}, \text{image}\}$. In line with the most prominent choices, we assume BPE for text (Gage, 1994) and VQ for images (Esser et al., 2021; Van Den Oord et al., 2017). Alice's goal is *out-of-model, generation-time, zero-bit watermarking* (see Sec. 2), i.e., embedding a later detectable watermark in *all* outputs of $\mathcal{M}$, without modifying the model's weights. We assume that Bob has only black-box access to $\mathcal{M}$, and no access to any $\mathcal{T}_m$ or $\mathcal{D}_m$.

Table 1: Average token match between 1000 image token sequences generated with TAMING (see Sec. 4) and their re-tokenized versions, which may also undergo image transformations before re-tokenization.

| Original | Blur $ksz = 9$ | Noise $\sigma = 0.1$ | JPEG $Q = 25$ | Brighten $2\times$ | Rotate $10°$ | Flip $\leftrightarrow$ | Crop $0.75$ |
|---|---|---|---|---|---|---|---|
| 0.66 | 0.26 | 0.17 | 0.31 | 0.11 | 0.02 | 0.01 | 0.01 |

**Adapting LLM watermarking.** When $\mathcal{M}$ generates text, we directly apply KGW (Sec. 2) with $h = 1$. For images, using a fixed split ($h = 0$), known to make watermarks insecure for text, i.e., easy to reverse-engineer (Jovanović et al., 2024; Kirchenbauer et al., 2024; Zhang et al., 2024c), may in our case be a more viable choice due to the opacity of the VQ tokenizers. We thus explore $h \in \{0, 1\}$ in Sec. 4, with more variants in App. F. Another degree of freedom is the choice of watermark context—we did not find exploiting the 2D structure of images to be beneficial, despite the intuition that spatially close contexts benefit robustness.

**Detection.** Given samples $x^{(i)}$ of varying modalities that Alice suspects were generated by $\mathcal{M}$ (e.g., a post on a breed of hen as in Fig. 1), she can apply Eq. (2) in a unified way. We first tokenize each $x^{(i)}$ to $s^{(i)}$ of length $T^{(i)}$ tokens, and score it using the corresponding $h^{(i)}$ to obtain a score $S^{(i)}$. We next sum all $S^{(i)}, T^{(i)}$, and $h^{(i)}$, and deduplicate scored (context, token) pairs across all samples to preserve statistical soundness (Fernandez et al., 2023a; Jovanović et al., 2025; Kirchenbauer et al., 2023; Sander et al., 2024). Then, we apply Eq. (2) to obtain a single p-value. Notably, the same $\gamma$ must be used across all modalities. Alice may reject $\mathcal{H}_0$ (flag content as watermarked) if the p-value is below the desired false positive rate (FPR). In Sec. 4.3 we investigate the benefits of jointly watermarking multiple modalities, and discuss the involved tradeoffs.

In Fig. 2, we visualize the watermark on an image generated with TAMING (Esser et al., 2021), by applying it only on the second half of generated tokens. As we later confirm in Sec. 4, our watermark has high power (low p-value), while imperceptibly modifying images by altering semantics.

**Challenge: reverse cycle-consistency (RCC).** The tokens $s$ shown in Fig. 2 as input to the detector are those generated by the autoregressive model, which is not realistic. In practice, to apply the detector to a sample $x'$, Alice must first tokenize it as $s' = \mathcal{T}_m(x')$. If tokens $s'$ significantly differ from $s$, the watermark may be lost. To quantify this, we define the *token match* as:

$$\text{TM}(s, s') = \frac{1}{T} \sum_{i=1}^{T} \mathbb{1}(s_i = s_i'), \quad (3)$$

where $s' = \mathcal{T}_m(\mathcal{D}_m(s))$. We say that *reverse cycle-consistency (RCC)* holds if $\text{TM}(s, s') \approx 1$.

RCC is not guaranteed even in text, despite BPE tokenizers ensuring *forward cycle-consistency (FCC)*, i.e.,

$\mathcal{D}_{\text{text}}(\mathcal{T}_{\text{text}}(x)) = x$. Namely, two tokens may be merged if the resulting token also exists in the vocabulary (see App. C for details and examples). Still, RCC largely holds in practice, evidenced by no prior LLM watermarking citing related challenges. We confirm this experimentally: across 1000 completions from LLAMA3.1-8B-INSTRUCT, average token match was 0.995.

**RCC in image generation.** We repeat this experiment on image models, presenting the results in Table 1 (full details in App. E). We also consider the case where images undergo transformations ($x \rightarrow a(x)$) before re-tokenization. Without transformations (*Original*), RCC is already weaker than expected with TM = 0.66. As Sec. 4 will show, this often suffices under ideal conditions (e.g., our example in Fig. 2 has a p-value of $10^{-9}$ after re-tokenization). However, common valuemetric transformations (blur, noise, JPEG, brighten) lower TM (e.g., to 0.31 for JPEG with $Q = 25$), and geometric ones (rotate, flip, crop) cause a further drop to almost 0. Two key factors explain this behavior. First, neural image tokenizers are trained for FCC, not RCC. Their training data does not include detokenized samples, which often lie off-manifold. Second, spatial sensitivity of the tokenizer allows semantic-preserving edits to easily alter most tokens. We next show how to mitigate this.

### 3.1 Finetuning for Reverse Cycle-consistency

We propose a finetuning procedure (illustrated in Fig. 3) that improves RCC in image tokenizers. Recall the VQ components (Sec. 2): encoder $E$, quantizer $Q_C$ with codebook $C$, and decoder $D$. Let $D_0, E_0$ be the original weights of $D$ and $E$. To avoid costly retraining of $\mathcal{M}$, we must keep $(E, Q_C, C)$ fixed; otherwise, we risk modifying the codebook semantics (directly or by changing how images are encoded by $E$), which harms the autoregressive model. Thus, we propose to only finetune $D$ and an encoder *replica* $E'$ (initialized to $E_0$). $E'$ is used only for watermark detection, while the original $E$ may be used to condition $\mathcal{M}$ on images. Unlike usual VQ training that promotes FCC, we optimize RCC: we aim to learn a decoder $D$ whose outputs $E'$ can reliably invert.

**Finetuning objectives.** We first precompute tokenizations $s$ from a set of images, which we use as our training data. We encourage RCC by minimizing the following loss:

$$\mathcal{L}_{\text{RCC}}(s) = \mathbb{E}_{a \sim \mathcal{A}} \|\hat{z} - E'(a(D(\hat{z})))\|_2^2. \quad (4)$$

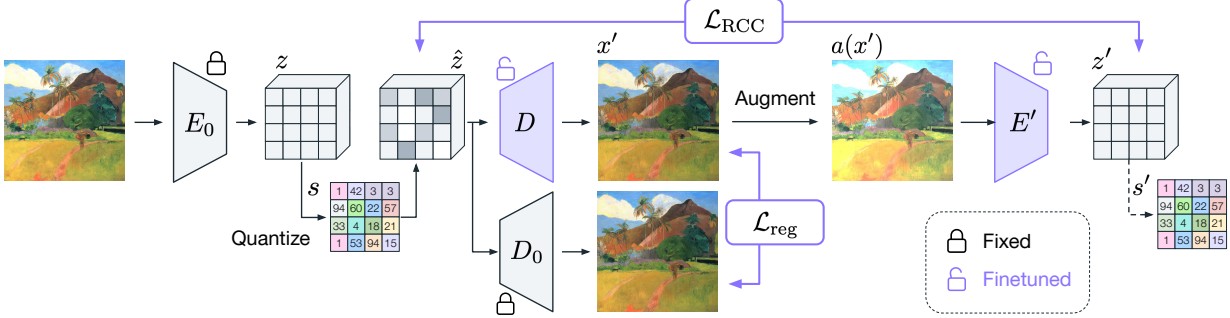

Figure 3: A replica $E'$ of the encoder and the decoder $D$ are jointly trained to improve reverse-cycle consistency, i.e., make $E'(D(\hat{z}))$ close to $\hat{z}$ for most generations of the autoregressive model $\mathcal{M}$, even under transformations.

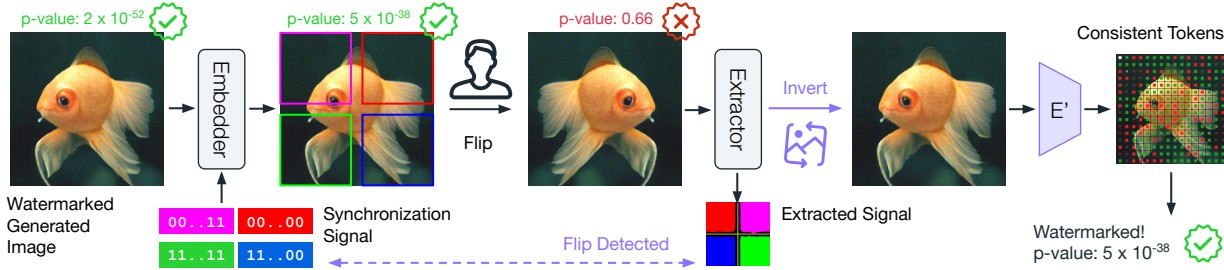

Figure 4: Watermark synchronization. Localized messages are embedded into a generated watermarked image and later used to discover the unknown transformation and revert it, which recovers the original watermark.

Its goal is to match the original *hard latents* $\hat{z} = C_s$ to *soft latents* obtained after detokenization and encoding using $E'$. To ensure RCC holds robustly even under transformations, we uniformly sample an augmentation $a \sim \mathcal{A}$ with preset probability $p_{\text{aug}}$ in each training step, or set it to identity otherwise. Our augmentations set $\mathcal{A}$ includes valuemetric (brighten, contrast, JPEG) and weak geometric transformations (e.g. $\pm 1°$ rotation), with strength ramped up over training (see App. E).

To retain decoder quality we introduce a regularization that keeps $D$ close to $D_0$ via a mixture of MSE and LPIPS perceptual loss (Zhang et al., 2018):

$$\mathcal{L}_{\text{reg}}(s) = \|D(\hat{z}) - D_0(\hat{z})\|_2^2 + \mathcal{L}_{\text{LPIPS}}(D(\hat{z}), D_0(\hat{z})). \tag{5}$$

We found this sufficient as a quality constraint and easier to train compared to targeting the original images or using adversarial discriminators. With tradeoff parameter $\lambda$, we jointly train $D$ and $E'$ to minimize:

$$\mathcal{L}(s) = \mathcal{L}_{\text{RCC}}(s) + \lambda \cdot \mathcal{L}_{\text{reg}}(s). \tag{6}$$

In Sec. 4 we show that this greatly boosts RCC and watermark robustness, especially against valuemetric attacks, neural compression, and diffusion purification.

### 3.2 Post-hoc Watermark Synchronization

Semantic-preserving transformations such as flips easily change image tokenization as each token loosely corresponds to a local image patch. Therefore, RCC finetuning alone cannot recover the watermark. One could run the watermark detector on multiple transformed images (rescaled, rotated, etc.), but this is costly and significantly inflates false positives as noted in prior work (Kirchenbauer et al., 2023).

**Localization as synchronization.** To address this in a more practical way, we repurpose localized watermarks (see Sec. 2) as a synchronization signal. More precisely, we locally embed a fixed set of messages whose detection estimates the applied transform, which we then aim to invert before applying the original watermark detector. Detecting this signal could in principle be taken as evidence that the image is watermarked, as in some of the prior post-hoc watermarking schemes that explore synchronization (Guo et al., 2023; Luo et al., 2022). However, applying the original watermark detector is still necessary to obtain theoretically-grounded p-values that can be combined with other samples across modalities as described above. Moreover, as we will see in Sec. 4, post-hoc watermarks are generally much more brittle to adversarial purification compared to the approach we propose. To not degrade original performance, our watermark should be robust to the addition of this signal, which we verify in Sec. 4.

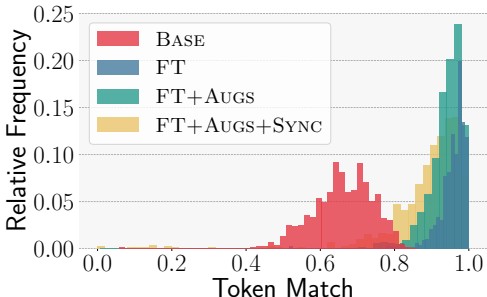 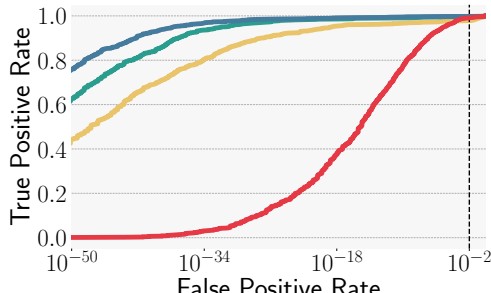

Figure 5: **Left**: Finetuning improves token match (Eq. (3)) between original and re-tokenized image tokens. **Right**: All variants achieve TPR ≈1 at FPR of 1%. Finetuning further boosts detection in low-FPR settings.

**Reverting transformations.** In Fig. 4, we show four 32-bit synchronization messages $\{0^{32}, 0^{16}1^{16}, 1^{16}0^{16}, 1^{32}\}$ embedded via the method of Sander et al. (2025) into the four image quadrants. We observe that adding the synchronization signal does not significantly degrade the original watermark ($p = 5 \cdot 10^{-38}$). However, a horizontal flip shuffles tokens and breaks detection ($p = 0.66$). To identify this, we apply an algorithm that searches over a grid of rotation angles, and for each fits the best axis-aligned pair of orthogonal lines that separate the four messages. In our example, this detects that a flip was applied and restores $p = 5 \cdot 10^{-38}$. Note that we consider crops that are followed by upscaling to the respective model's original generation size—our synchronization reverts this by applying downscaling and padding. A more detailed description and additional examples are deferred to App. D.

Next, we empirically show that synchronization enhances geometric robustness, complementing RCC finetuning. This step is further aided by the use of small geometric augmentations during RCC finetuning, as they effectively compensate for error in our transformation estimates.

## 4 Experimental Evaluation

In Sec. 4.1, we measure the effect of finetuning (Sec. 3.1) and synchronization (Sec. 3.2) on RCC, quality, and the power of our watermark. Sec. 4.2 studies robustness, while Sec. 4.3 explores joint watermarking of text and images. Additional details and results are given in App. E and App. F.

**Setup.** We use the class-conditional ImageNet transformer from Esser et al. (2021) at resolution $256 \times 256$ with a VQ-GAN tokenizer with $|V| = 16384$ and $f = 16$ (TAMING) and the 7B mixed-modal CHAMELEON (Chameleon Team, 2024) that can generate interleaved text and $512 \times 512$ images. We always generate 1000 samples (10 ImageNet classes for TAMING, 1000 COCO prompts for CHAMELEON), and evaluate 4 variants of our method: BASE, which uses original models and tokenizers, FT and FT+AUGS, which apply the same watermark after RCC finetuning (Sec. 3.1) without and with augmentations in

training, respectively, and FT+AUGS+SYNC, which also uses our watermark synchronization (Sec. 3.2). We use $\delta = 2$, $\gamma = 0.25$ in all experiments, $h = 1$ for TAMING and CHAMELEON on text, and $h = 0$ for CHAMELEON on images. We finetune models on tokens derived from 50,000 ImageNet training samples for 10 epochs (2h on 16 V100 for TAMING and 2.5h on 8 H200 for CHAMELEON).

### 4.1 RCC, Watermark Power and Quality

The key question raised in Sec. 3 is if finetuning can alleviate the lack of reverse cycle-consistency (RCC) in image tokenizers, and in turn improve watermark power.

**Finetuning improves RCC.** We generate 1000 class-conditioned ImageNet samples with TAMING using each of our four variants, and measure token match (TM, Eq. (3)) between the generated tokens and those obtained by re-tokenizing the image. In our results in Fig. 5 (left), we observe that TM is consistently below 0.8, while for all finetuned variants it is generally *above* 0.8. This demonstrates that finetuning is successful in improving RCC. AUGS and SYNC slightly reduce TM on unmodified images, but significantly increase robustness (see Sec. 4.2).

**Finetuning improves watermark power.** In Fig. 5 (right), we report the true positive rate (TPR) of the watermark detector for different false positive rates (FPR). The BASE variant already has practically viable power, with TPR of ≈1 at FPR of $10^{-2}$ (dashed line), the setting considered in prior work (Ci et al., 2024; Dathathri et al., 2024; Wen et al., 2023; Zhao et al., 2023). However, RCC gains directly translate to improvements in watermark power: for all 3 variants, the TPR at lower FPR settings is significantly higher.

**Watermarking does not harm generation quality.** To measure quality, we compute FID (Heusel et al., 2017) on 50,000 generated samples (50 per ImageNet-1K class) for all variants. We find that none of BASE, FT, and FT+AUGS have FID above 16.7, the FID of *unwatermarked* BASE. This confirms that our finetuning preserves generation quality. The FID of FT+AUGS+SYNC is 17.3, a minor increase inherited from the localized watermark used for synchro-

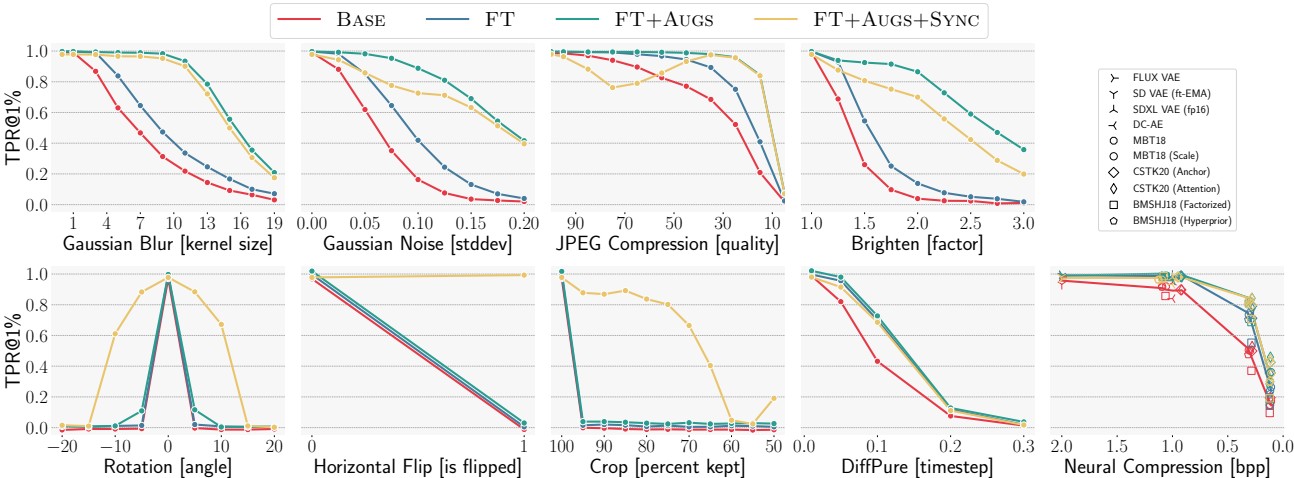

Figure 6: **Top:** RCC finetuning improves robustness to valuemetric transformations. **Bottom Left:** Watermark synchronization unlocks robustness to geometric transformations. **Bottom Right:** Our watermark is also fairly robust to realistic strengths of diffusion purification (Nie et al., 2022; Saberi et al., 2024) and neural compression (Ballé et al., 2018; Chen et al., 2024; Cheng et al., 2020; Labs, 2024; Minnen et al., 2018; Podell et al., 2024; Rombach et al., 2022).

nization. We complement this with qualitative samples in App. G. Repeating this on CHAMELEON (App. F) leads to the same conclusion.

### 4.2 Watermark Robustness

An important requirement for a generative model watermark is robustness to common domain-specific transformations, as well as to removal attacks, which have shown to be effective against other watermarks (An et al., 2024; Fernandez et al., 2023b; Saberi et al., 2024). To evaluate this, in Fig. 6 we report the watermark TPR for a fixed FPR of $1\%$ on a range of transformations of different strength, in the same setting as in Fig. 5, using TAMING. We summarize these and corresponding results for CHAMELEON in Table 2, where, as in prior work (Wen et al., 2023), we average TPR over a set of (transformation, parameter) pairs, detailed in App. E.

**Finetuning enables valuemetric and attack robustness.** In Fig. 6, we see that the watermark is fragile to valuemetric and geometric transformations when used on BASE. When we use it on FT+AUGS, robustness to valuemetric transformations greatly improves, validating our focus on RCC finetuning. Surprisingly, finetuning also improves robustness to (i) neural compressors (Ballé et al., 2018; Cheng et al., 2020; Minnen et al., 2018) of different strengths (see details in App. E), including FLUX and SD VAEs (Chen et al., 2024; Labs, 2024; Podell et al., 2024; Rombach et al., 2022), and (ii) the challenging diffusion purification attack (Nie et al., 2022). We remark that high values such as $t = 0.3$ were found to excessively alter images, making this regime less relevant (Saberi et al., 2024). This effect is less pronounced for CHAMELEON in Table 2, where our watermark is already robust to these attacks even without RCC finetun-

ing, likely due to the detector scoring more tokens for larger images.

**Synchronization enables geometric robustness.** Geometric robustness (*bottom left* in Fig. 6) remains extremely low for both models, even for FT+AUGS. This is expected, as the design of autoregressive models makes it impossible to preserve token sequences under semantic changes such as flips. This motivated our synchronization layer (Sec. 3.2), which we observe significantly improves geometric robustness, for a minor drop in valuemetric robustness. This drop is the most pronounced for moderate transformation strengths, when a wrong estimate corrupts the tokens— fully preserving (for low strengths) or destroying (for high strengths) the synchronization signal does not impact results. Importantly, watermark power without transformations remains close to 1 even after synchronization. In Sec. 6, we propose several ways to improve this tradeoff in future work.

**Comparison to post-hoc methods.** As we noted in Sec. 1, no prior work targets watermarking of autoregressive models for images. Thus, in Table 2 we compare to several *post-hoc* methods (Bui et al., 2023a; Jia et al., 2021; Ma et al., 2022; Sander et al., 2025) applied on top of generated images (with the default sampling). While they are comparably or more robust than our watermark on valuemetric transformations, each post-hoc watermark is either fully removed by geometric ones or not highly robust to attacks (adversarial purification and neural compressors). More importantly, our watermark yields p-values grounded in the randomness of the detection process, with theoretical guarantees inherited from LLM watermarking (Fernandez et al., 2023a; Kirchenbauer et al., 2024; Zhao et al., 2023) and em-

Table 2: TPR at 1% FPR of our watermark and post-hoc baselines, under valuemetric (**Val.**) and geometric (**Geo.**) transformations, adversarial purification attacks (**Adv.**) and neural compression (**NC**). Scores (explained in App. E) below 0.6 are marked red. Finetuning and synchronization lead to a strong and robust watermark.

| | | TAMING (256 × 256) | | | | | CHAMELEON (512 × 512) | | | | |
|---|---|---|---|---|---|---|---|---|---|---|---|
| | | **None** | **Val.** | **Geo.** | **Adv.** | **NC** | **None** | **Val.** | **Geo.** | **Adv.** | **NC** |
| Ours | BASE | 0.99 | 0.26 | 0.01 | 0.43 | 0.48 | 0.98 | 0.50 | 0.02 | 0.80 | 0.82 |
| | FT | 1.00 | 0.45 | 0.01 | 0.70 | 0.71 | 0.99 | 0.53 | 0.03 | 0.85 | 0.87 |
| | FT+AUGS | 1.00 | 0.92 | 0.01 | 0.70 | 0.79 | 0.99 | 0.89 | 0.02 | 0.82 | 0.88 |
| | FT+AUGS+SYNC | 0.98 | 0.83 | 0.82 | 0.69 | 0.80 | 0.97 | 0.76 | 0.64 | 0.81 | 0.86 |
| Post-hoc | CIN | 1.00 | 0.96 | 0.00 | 0.03 | 0.02 | 1.00 | 0.99 | 0.00 | 0.14 | 0.16 |
| | MBRS | 1.00 | 0.98 | 0.02 | 0.36 | 0.31 | 1.00 | 0.99 | 0.02 | 0.27 | 0.56 |
| | TRUSTMARK | 1.00 | 0.98 | 0.75 | 0.40 | 0.86 | 1.00 | 0.97 | 0.74 | 0.64 | 0.99 |
| | WAM | 1.00 | 0.89 | 0.98 | 0.06 | 0.02 | 1.00 | 0.97 | 0.95 | 0.26 | 0.48 |

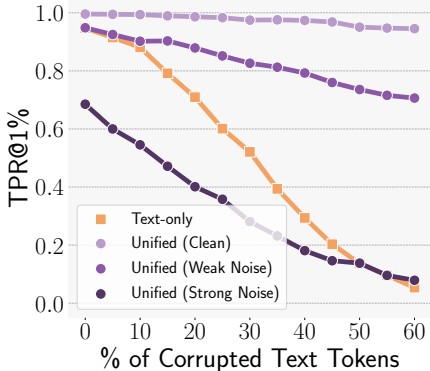

Figure 7: Joint watermark detection on text and image generations.

pirically validated in App. F. In contrast, post-hoc methods rely on neural extractors to recover message bits and may introduce bias in their theoretical p-value estimators (Fernandez et al., 2023b, App. B.5) and (San Roman et al., 2024, App. B). Finally, as a token-level generation-time method, our watermark is the only one able to watermark content via semantic modifications (see Fig. 2). In App. F we provide a comparison to generation-time watermarks for diffusion models, despite their inapplicability to our target models.

### 4.3 Joint Watermarking of Interleaved Modalities

Finally, we explore joint watermarking of multiple modalities generated by the same autoregressive model. Eq. (2) shows that scoring more equally watermarked tokens improves power. However, acquiring more tokens is not always possible—in such cases, jointly watermarking multiple modalities may be necessary to reliably detect the watermark. For example, consider that Alice aims to prove if an online article was generated by her model $\mathcal{M}$. To simulate this, we run CHAMELEON (FT+AUGS) in interleaved mode on 1000 prompts to produce text and an image, and we model attempts to conceal the use of $\mathcal{M}$ by randomly changing text tokens (a proxy for paraphrasing).

**Benefits of joint detection.** The orange line in Fig. 7 shows TPR at 1% FPR when only text is watermarked, quickly degrading with text corruption. As all suspect text is used, it is hard for Alice to improve detection in this scenario. However, if both text and the image were originally watermarked with our method, detection on combined tokens as described in Sec. 3 significantly boosts power (top purple, *Clean*). At 10% corruption TPR improves from ≈0.9 to 1.0, and stays above 0.94 even in the hardest case, where text-only TPR drops to ≈0. Alice gets a rigorous p-value, which would be hard if modalities were watermarked separately. As our method is robust to moderate image transformations, a similar trend holds when adding Gaussian noise with $\sigma = 0.1$ (middle purple, *Weak Noise*).

Importantly, there is a tradeoff—*integrating a weak watermarking signal can degrade detection*. We see this for $\sigma = 0.3$ (bottom purple, *Strong Noise*), where TPR drops below 0.6 at 10% corruption, i.e., text-only detection is preferable. In App. H we discuss further and show extended results. Inspired by this, in the following section we explore the extension of our method to additional modalities.

## 5 Extension to Additional Modalities: Audio Case Study

We here ask: *Can our approach be extended to other modalities?* We give a preliminary study on autoregressive audio generation (Borsos et al., 2022; Copet et al., 2024; Défossez et al., 2024; Nguyen et al., 2025; Zhang et al., 2023a; 2024b) focusing on MOSHI (Défossez et al., 2024), a transformer-based speech-text foundation model. We observe similar challenges and main results as for images, while noting several important differences. Complete details and full results are given in App. E and F.

**Audio tokenizer.** MOSHI's tokenizer (MIMI) relies on *residual* vector quantization (RVQ) (Défossez et al., 2022; Kumar et al., 2023; Lee et al., 2022; Zeghidour et al., 2022).

Table 3: TPR at 1% FPR and MOSNet (Lo et al., 2019) on 1000 audio samples generated with MOSHI with our watermark, under valuemetric (**Val.**) and time-frequency (**Time**) transformations, and neural compression (**NC**). MOSNet is 3.80 for unwatermarked generation.

|       |          | None | Val. | Time | NC  | MOSNet |
|-------|----------|------|------|------|-----|--------|
| Ours  | BASE     | 0.97 | 0.62 | 0.24 | 0.80 | 3.82  |
|       | FT       | 0.99 | 0.64 | 0.14 | 0.84 | 3.83  |
|       | FT+AUGS  | 0.99 | 0.80 | 0.24 | 0.86 | 3.73  |
| Post-hoc | AUDIOSEAL | 1.00 | 0.84 | 0.55 | 0.85 | 3.78 |

RVQ iteratively quantizes the residuals of the previous quantizer, such that $s_i = (s_i^1, .., s_i^K)$ for $K$ different codebooks $C^1, .., C^K$ ($K$ *streams*) Each token here represents $\approx 80$ ms.

**RCC in audio.** Défossez et al. (2024, Sec. 6.4) observe that the first stream is somewhat cycle-consistent, while this degrades for later ones. We obtain similar results and show that TM further worsens under transformations (highpass, speedup). For instance, we measure TM = 0.36 (original), 0.21 (highpass 500 Hz), 0.16 (1.1× speedup), averaged over all streams on 1000 generated sequences. This motivates adapting RCC finetuning (Sec. 3.1) and synchronization (Sec. 3.2) to audio.

**RCC finetuning.** To instantiate the finetuning procedure from Sec. 3.1 we make the following changes to Eqs. (4) and (5). We use the pre-projection soft latents as target since the quantization is done in a projected space (Kumar et al., 2023). We replace LPIPS with multi-resolution STFT loss (Yamamoto et al., 2020). Finally, during training we apply augmentations from a set $\mathcal{A}$ that includes audio-specific valuemetric edits (high/low/bandpass, gaussian/pink noise, etc.) as well as small (1-10 ms) time-frequency shifts.

**Synchronization.** In contrast to images, no localized audio embedder proved robust: the localization property of San Roman et al. (2024) is not precise enough under mild time-frequency edits. Future work on audio-specific localized watermarking could possibly help; our experiment here focuses on the influence of RCC finetuning.

**Watermarking multiple streams.** Early RVQ streams are more reverse cycle-consistent and thus more likely to preserve the watermark signal. However, limiting watermarking to one stream provides too few tokens for reliable statistical testing, significantly increasing p-values. On the other hand, watermarking all streams introduces noise due to the lack of RCC in later codes. Empirically, we find that watermarking the first four streams achieves a good balance.

**Experimental setting.** We train FT and FT+AUGS on VoxPopuli (Wang et al., 2021) such that final PESQ (Rix et al., 2001) is 4.3 w.r.t. BASE samples. We generate 12s watermarked audio samples with MOSHI using 1000 text

prompts generated by LLAMA3.1-8B-INSTRUCT and synthesized to audio with SEAMLESSV2 (Barrault et al., 2023). We use $h = 0, \delta = 2$ and watermark the first four audio streams. As in Sec. 4, we evaluate TPR at 1% FPR and quality, for which we use MOSNet (Défossez et al., 2024).

**Results.** We present the results in Table 3. As for images, we do not observe notable quality degradation due to watermarking. We find that BASE already has nonzero time-frequency robustness, likely due to non-semantic streams being used to carry the watermark. Finetuning in this case boosts valuemetric robustness, but, interestingly, impairs time-frequency RCC, which is recovered by FT+AUGS. We hypothesize that this drop is due to catastrophic forgetting (Kirkpatrick et al., 2017) as the model learns to detokenize the audio in a way that is not robust to time-frequency transformations. This suggests that augmentations are a key component of finetuning, matching our results on images. While we are not aware of audio equivalents of diffusion purification used in Sec. 4, we note robustness to neural compression (we use DAC (Kumar et al., 2023) and EnCodec (Défossez et al., 2022)) comparable to post-hoc AUDIOSEAL (San Roman et al., 2024), even though in contrast to AUDIOSEAL we do not explicitly train against EnCodec.

## 6 Conclusion and Limitations

Our work successfully applies watermarking to the previously unexplored setting of autoregressive image generation, addressing low reverse cycle-consistency (RCC) through a custom finetuning stage and a synchronization layer. Experiments demonstrate the power, robustness, and practicality of our watermark across a range of settings. By broadening the scope of watermarking, we believe this work takes an important step towards more reliable content provenance. We further discuss societal and environmental impact in App. A.

**Limitations.** Our method's scope could be extended further. As noted in Sec. 2, we target the most prominent models that tokenize images via VQ. Our method does not apply to models using continuous representations or hybrids like autoregressive-diffusion models (Fan et al., 2024; Li et al., 2024; Ma et al., 2024; Zhou et al., 2024). Another dimension is modality: we present initial audio experiments in Sec. 5, but this could be extended further. Next, our synchronization relies on off-the-shelf localized watermarks, which are suboptimal as they embed arbitrary patterns. A more principled approach would be to train a bespoke synchronization layer for embedding a fixed pattern, integrated with RCC finetuning for added robustness. Finally, our method is not robust to combined removal attacks (to disrupt synchronization) and geometric attacks (to decrease token match)—to the best of our knowledge, this attack would also break most other contemporary watermarks. We leave this question open for future work.

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

# Appendix

## A   Ethical Statement

### A.1   Societal Impact

Watermarking in general improves the traceability of content, be it AI-generated or not. It can have positive consequences, for example when it is used to trace the origin of fake news or to protect intellectual property. This traceability can also have negative consequences, for example when it is used to trace political opponents in authoritarian regimes or whistleblowers in secretive companies. Besides, it is not clear how to disclose watermark detection results, which may foster a closed ecosystem of detection tools. It may also exacerbate misinformation by placing undue emphasis on content that is either not detected, generated by unknown models, or authentic but used out of context. We however believe that the benefits of watermarking outweigh the risks, and that the development of robust watermarking methods is a positive step for our society.

### A.2   Environmental impact

The cost of the experiments and of model training is high, though order of magnitude less than training the generative models themselves. Finetuning the image tokenizer as done in the paper takes $\leq 32$ GPU-hours. We also roughly estimate that the number of GPU-days used for running all our experiments is around 500, i.e., 12k GPU-hours. This amounts to total emissions in the order of 1 ton of $CO_2$eq[1]. Estimations are conducted using the Machine Learning Impact Calculator presented by Lacoste et al. (2019). We do not consider in this approximation: memory storage, CPU-hours, production cost of GPUs/ CPUs, etc.

## B   Technical Details of LLM Watermarking

We here more thoroughly introduce LLM watermarking, following the notation in Sec. 2.

**Generation.**   We consider an autoregressive model $\mathcal{M}$ generating a sequence of tokens $s = (s_1, s_2, \ldots, s_T)$, where each token $s_t$ is sampled from a probability distribution conditioned on the previous tokens $p(s_t | s_{<t})$. In practice, the model outputs a vector of logits $\ell \in \mathbb{R}^{|V|}$, where $V$ is the vocabulary (which we can assume in the most general case can contain text, audio or image tokens), which is transformed into a probability distribution $p = \mathrm{softmax}(\ell/\tau)$, with $\tau$ being a temperature parameter.

The *watermark embedding* modifies the token selection process using a secret key $\xi$. A cryptographic hash function takes as input $h$ previous tokens $(s_{t-h}, \ldots, s_{t-1})$ (the context window) and the secret key $\xi$, producing a seed for a random number generator (RNG) that influences the selection of the next token $s_t$.

Two prominent watermarking approaches are:

- Kirchenbauer et al. (2023)'s method (KGW) uses RNG to randomly partition the vocabulary $V$ into a greenlist $\mathcal{G}_t$ and a redlist $\mathcal{R}_t$, where $\mathcal{G}_t$ contains a proportion $\gamma$ of the vocabulary. The logit of each token in the greenlist is increased by $\delta > 0$, effectively boosting the probability of selecting tokens from the greenlist.

- Aaronson & Kirchner (2023)'s method uses a different approach based on the RNG to sample secret values for each token. Although we do not present it in this work for simplicity, it could have been adapted in the same way to watermark autoregressive models.

**Detection.**   For the KGW method that we focus on, the *watermark detection* process analyzes a token sequence $s$ and computes a score $S$ based on the count of green tokens:

$$S = \sum_{t=h+1}^{T} \mathbb{1}(s_t \in \mathcal{G}_t), \tag{7}$$

where $\mathcal{G}_t$ is the greenlist for position $t$, which depends on the $h$ preceding tokens and the secret key $\xi$.

---

[1]Using a default grid, we compute 250W $\times$ 12000h = 3000 kWh $\times$ 0.3 kg eq. CO2/kWh = 900 kg eq. CO2

**Statistical hypothesis testing.** Detection uses a statistical hypothesis test distinguishing between $\mathcal{H}_0$: "the sequence is not watermarked with secret key $\xi$" and the alternative $\mathcal{H}_1$: "the sequence was generated with a watermark with secret key $\xi$." Previous approaches, such as those by Kirchenbauer et al. (2023) and Aaronson & Kirchner (2023), relied on a $Z$-test to compare the count of green tokens $S$ to its expected value under the null hypothesis $\mathcal{H}_0$. In this work we instead adopt an exact test (Fernandez et al., 2023a), which is more accurate, especially for short sequences.

Under $\mathcal{H}_0$, $S$ follows a binomial distribution $\mathcal{B}$ with parameters $(T - h)$ and $\gamma$, where $\gamma$ is the expected proportion of green tokens, $T$ is the total number of tokens, and $h$ is the size of the watermark context window. The p-value determines the likelihood of observing a score as extreme as $S$ under $\mathcal{H}_0$, and is calculated as:

$$\text{p-value}(S, T, h, \gamma) = \text{Prob}\left(X \geq S \mid X \sim \mathcal{B}(T - h, \gamma)\right) = I_\gamma(S, T - h - S + 1), \tag{8}$$

where $I_x(a, b)$ is the regularized incomplete Beta function.

Sequences are flagged as watermarked if the p-value falls below the desired false positive rate.

**Main parameters.** The main parameters of the watermarking method are the context window size $h$, the watermark strength factor $\delta$ and the proportion of green tokens $\gamma$.

The context window size $h$ determines how many previous tokens determine the greenlist. A smaller $h$ increases robustness against text modifications but may bias generation as the same hash is used more frequently. It typically reduces security since recurring greenlists make the watermark easier to spoof (Jovanović et al., 2024; Zhang et al., 2024c). When $h = 0$, the RNG seed depends solely on the secret key $\xi$, creating fixed green/red lists for all tokens. For non-text tokens, we hypothesize that $h = 0$ maintains security since tokenizer access is restricted and image tokenizers have more degrees of freedom than text ones.

The watermark strength factor $\delta$ determines the amount by which the logits of green tokens are boosted. A higher $\delta$ increases the robustness of the watermark, but also increases the risk of generating low-quality text/images. It is tuned for every model and application.

The proportion of green tokens $\gamma$ affects both detection sensitivity and generation quality. With low $\delta$, a smaller $\gamma$ reduces green token selection during generation, resulting in lower watermark power. With high $\delta$, it restricts token choice and may lower output quality. During detection, lower $\gamma$ values yield more significant p-values since green tokens are less likely to appear by chance (Kirchenbauer et al., 2023). At fixed watermark power, higher $\gamma$ distributes the watermark evenly, while lower values concentrate it on fewer tokens. We set $\gamma$ to $0.25$ in our experiments, as it is a common choice in the literature (Aaronson & Kirchner, 2023; Kirchenbauer et al., 2023) and consistently yields good results in our experiments.

## C More on Reverse Cycle-consistency

In this section, we elaborate on the case of text tokenizers not being perfectly reverse cycle-consistent (RCC), discuss audio tokenizers and our experiment measuring RCC in this setting, expanding on Sec. 5, and discuss related topics studied in prior work.

### C.1 RCC in Text Tokenizers

In BPE tokenizers, the vocabulary is initialized with all characters in the training set, and common character pairs are iteratively merged and added to the vocabulary until the predefined size is reached. Tokenization is performed greedily from left to right, by always selecting the longest possible token from $V$. Detokenization is simply performed by a lookup into $V$.

**RCC can be violated.** Text tokenizers are not immune to the RCC issue. For example, consider the following subset of the GPT-4O tokenizer: {cons: 9673, istent: 20908, consistent: 173878}. Due to the greedy property of BPE tokenizers, $\mathcal{D}_{\text{text}}$ is guaranteed to always invert $\mathcal{T}_{\text{text}}$, e.g., $\mathcal{D}_{\text{text}}(\mathcal{T}_{\text{text}}(\texttt{consistent})) = \mathcal{D}_{\text{text}}([173878]) = \texttt{consistent}$, guaranteeing *forward cycle-consistency (FCC)*. In contrast, *reverse cycle-consistency (RCC)*, necessary for a strong watermark, may be violated, e.g., $\mathcal{T}_{\text{text}}(\mathcal{D}_{\text{text}}([9673, 20908])) = \mathcal{T}_{\text{text}}(\texttt{consistent}) = [173878]$. RCC approximately holds for text tokenizers in practice: while it is also a prerequisite for successful watermarking in text, no prior art has highlighted this as a hurdle. Some works have even shown that adversaries learning about the watermark can still be successful even if they use a different tokenizer (Jovanović et al., 2024), which is only possible if the tokens match across tokenizers.

Table 4: Token Match (TM) across different streams for 1k sequences, where audios are subject to various transformations before re-tokenization. Sequences of tokens are generated either as reconstructions of 10-seconds VoxPopuli audios with the MIMI tokenizer, or by the MOSHI model with audio prompts (described in App. E.3).

| | MIMI tokenizer | | | | | | | | | MOSHI | | | | | | | | |
|---|---|---|---|---|---|---|---|---|---|---|---|---|---|---|---|---|---|---|
| | 1 | 2 | 3 | 4 | 5 | 6 | 7 | 8 | Avg. | 1 | 2 | 3 | 4 | 5 | 6 | 7 | 8 | Avg. |
| Identity | 0.56 | 0.31 | 0.21 | 0.22 | 0.18 | 0.20 | 0.18 | 0.16 | 0.25 | 0.60 | 0.55 | 0.38 | 0.28 | 0.26 | 0.26 | 0.24 | 0.25 | 0.35 |
| *Transformations* | | | | | | | | | | | | | | | | | | |
| Lowpass 3 kHz | 0.38 | 0.15 | 0.15 | 0.17 | 0.14 | 0.16 | 0.15 | 0.12 | 0.18 | 0.50 | 0.39 | 0.30 | 0.21 | 0.19 | 0.21 | 0.20 | 0.21 | 0.28 |
| Noise 0.001 | 0.50 | 0.33 | 0.19 | 0.20 | 0.17 | 0.18 | 0.17 | 0.14 | 0.24 | 0.51 | 0.34 | 0.19 | 0.18 | 0.17 | 0.18 | 0.18 | 0.13 | 0.23 |
| MP3 16 kbps | 0.44 | 0.19 | 0.16 | 0.18 | 0.15 | 0.17 | 0.16 | 0.13 | 0.20 | 0.54 | 0.41 | 0.29 | 0.20 | 0.18 | 0.20 | 0.20 | 0.21 | 0.28 |
| Encodec | 0.24 | 0.13 | 0.10 | 0.10 | 0.07 | 0.07 | 0.05 | 0.04 | 0.10 | 0.28 | 0.38 | 0.24 | 0.16 | 0.14 | 0.15 | 0.11 | 0.15 | 0.20 |
| Speed ×1.25 | 0.02 | 0.01 | 0.01 | 0.00 | 0.00 | 0.00 | 0.00 | 0.00 | 0.01 | 0.04 | 0.11 | 0.08 | 0.05 | 0.05 | 0.06 | 0.04 | 0.07 | 0.06 |
| Crop (90% kept) | 0.03 | 0.01 | 0.01 | 0.01 | 0.01 | 0.01 | 0.01 | 0.00 | 0.01 | 0.08 | 0.15 | 0.11 | 0.07 | 0.07 | 0.08 | 0.05 | 0.09 | 0.09 |

**Experiment.**    As discussed in the main text, we confirmed this experimentally. We used LLAMA3.1-8B-INSTRUCT to generate 1000 answers to prompts from the Open Assistant dataset (Köpf et al., 2023). We then compared the generated sequences of tokens with the re-tokenized sequences. Specifically, we took the token IDs from the model's generation, detokenized them to text, then re-tokenized this text and computed the Levenshtein distance between the original and the new token sequence. Our results showed that the average token match is 99.5%, confirming that text tokenizers exhibit very high reverse cycle-consistency in practice.

### C.2  RCC in Audio Tokenizers

A study of the RCC issue in MOSHI's tokenizer is already given by the authors (Défossez et al., 2024) (called *idempotence* in their paper). We however observe some differences in our study, as well as other key findings, such as the effect of augmentations, that we summarized in Sec. 5, and that we discuss in more detail in the following.

**RVQ tokenizer.**    As a reminder, MOSHI's tokenizer (MIMI) utilizes *residual vector quantization* (RVQ) (Défossez et al., 2022; Kumar et al., 2023; Lee et al., 2022; Zeghidour et al., 2022). In RVQ, the quantization process happens iteratively, where each step quantizes the residual error from the previous quantization. Formally, for each step $i$, representing an audio frame of 1920 samples, the tokenization results in a sequence of tokens $(s_i^1, .., s_i^K)$ corresponding to $K$ different codebooks $C^1, .., C^K$ (referred to as $K$ *streams*). Each token represents approximately 80 ms of audio. The first token (or stream) is referred to as *semantic*, because there is a distillation loss during training with a non-causal model that encourages this first codebook to capture the most semantically relevant information. Défossez et al. (2024) note that this semantic token exhibits higher cycle-consistency compared to later streams, which are assumed to progressively capture more fine-grained details, and to be less consistent.

**Experimental setup.**    We measure Token Match (TM) for sequences either (a) generated as reconstructions of 10-seconds audios from VoxPopuli with the MIMI tokenizer, or (b) generated by the MOSHI model, as described in App. E.3. This corresponds to ≈125 time-steps for both cases, so ≈1,000 audio tokens (counting all the streams). The audio is subjected to various transformations before re-tokenization, which include the three categories: valuemetric (lowpass filtering at 3kHz, addition of strong Gaussian noise at 0.01 amplitude), temporal-frequency (speed modification by 1.25x, cropping 90% of the original audio), and compression-based (MP3 compression at 16kbps, EnCodec compression).

**Results.**    Table 4 presents the results that supplements the study by Défossez et al. (2024). Notably, their study only focuses on pre-existing audio sequences, while we also include the case of generated sequences, which behave differently. For instance, the second stream sometimes shows higher consistency than other streams contradicting the expectation that only the first (semantic) stream could maintain high consistency. Different augmentations affect streams differently: e.g., lowpass has less impact on streams 2 and 3 compared to strong noise addition, while EnCodec strongly decreases TM of the first stream. Importantly, temporal-frequency augmentations (speed, cropping) reduce TM less dramatically for MOSHI (0.04-0.15) than would be expected given our image watermarking results where such transformations typically yield near-zero consistency. This multi-stream aspect presents challenges and opportunities for watermarking: while complicating consistency analysis, it enables potential development of more sophisticated techniques leveraging complementary properties across streams.

### C.3 Related Concepts

There are several concepts related to RCC that were studied in prior work.

**Codec idempotence.** In the context of codecs a relevant property is *codec idempotence* (Kim et al., 2020; O'Reilly et al., 2025; Zhang et al., 2023b). Directly applying the mathematical definition of idempotence to our case, we let $f(\cdot)$ denote the encoder-decoder pair (e.g., $f$ compresses an image to a JPEG file and then decompresses it back to pixels), and say that a codec is idempotent if it satisfies:

$$f(f(x)) = f(x). \tag{9}$$

This property is naturally of interest to codecs. While we assume that JPEG compressing an image is lossy (i.e., we do not expect $f(x) = x$), we want our codec to not further degrade image quality on successive applications, which can commonly occur in practice (i.e., $f(f(x)) = f(x)$). This is in stark contrast with the notion of RCC relevant to our work. In our case, $x$ are the tokens, while $f(\cdot)$ is the detokenization followed by the tokenization. The first application of $f(\cdot)$ is crucial for us: as explained above, we require $f(x) \approx x$ as otherwise the watermark that was present in the tokens of $x$ is lost. $f(f(x))$, i.e., re-tokenizing the image several times successively, on the other hand is not of particular interest in this case, thus idempotence is not an important concern.

**Consistency of tokenizers.** Another related concern is the *consistency of tokenizers* (Liu et al., 2024b; Sun et al., 2023). Intuitively a tokenizer is *consistent* if the tokenization of a particular string (assuming the text domain) does not change depending on the surrounding context. While the cited works show that this is a desirable property, it is not as relevant to our motivation of preserving the watermark as RCC. In particular, for generated token sequence $x$, if RCC is satisfied the watermark will be entirely preserved, even if the tokenization was context-dependent. This may be a concern in the context of various attacks: for example, infilling a part of the image before re-tokenization should ideally change only the tokens corresponding to the infilled part, and not the entire image, which may happen depending on the setup of the convolutions in the tokenizer. We do not explore this angle as part of this work.

**Cycle-consistency in other contexts.** Finally, a line of works studies cycle-consistency in various generative models (Huang et al., 2020a; Zhu et al., 2017), most commonly in the context of style transfer: a single *cycle* is the translation from a style $A$ to a style $B$ and back to $A$, and cycle-consistency can in this case be beneficial as a constraint for the model. Finally, Teng & Choromanska (2019) explicitly parametrize encoder-decoder pairs to be inverses of each other and Huang et al. (2020b) study cycle-consistency in the context of disentangled representations.

## D More on Watermark Synchronization

In App. D.1 we provide a more detailed description of our watermark synchronization layer (Sec. 3.2), and show additional examples. In App. D.2 we describe our attempt to use AudioSeal (San Roman et al., 2024) for synchronization in audio.

### D.1 Image Synchronization Details

We remark that the problem of watermark synchronization was studied in the past, before the advent of generative models, in the context of digital watermarking. These works suggest approaches such as multiple testing with a carefully controlled number of tests to avoid the false positive rate increase we mentioned in Sec. 3.2 (Hartung et al., 2000), or similarly to us, embedding a synchronization pattern in addition to the original watermarking pattern to revert the transformation (Csurka et al., 1999; Pereira et al., 1999; Tirkel et al., 1998). As noted above, we are aware of two works that study this in the context of post-hoc generative model watermarks (Guo et al., 2023; Luo et al., 2022), however their code is not publicly available.

**Full algorithm description.** Our algorithm consists of two main procedures: embedding a synchronization pattern into the generated and decoded image, and estimating the transformation from an incoming image where previously the watermark and the synchronization pattern were embedded. We assume access to a localized watermark module $L$ that can embed a different message in every pixel of an image, and recover the probability that each pixel has the watermark along with the most probable message in it. As noted above, we instantiate this using Sander et al. (2025).

To embed the synchronization pattern, we use four 32-bit messages $\{m_1 = 0^{32}, m_2 = 0^{16}1^{16}, m_3 = 1^{16}0^{16}, m_4 = 1^{32}\}$. For each message, the corresponding mask is one of the quadrants (as in Fig. 4), where given parameter $\mu$, we keep a horizontal and a vertical strip of width $\mu$ pixels in the middle of the image free of messages (we use $\mu = 18$ for TAMING and $\mu = 36$ for CHAMELEON as we work with images of twice the resolution). Using this mask, we embed the pattern using $L$.

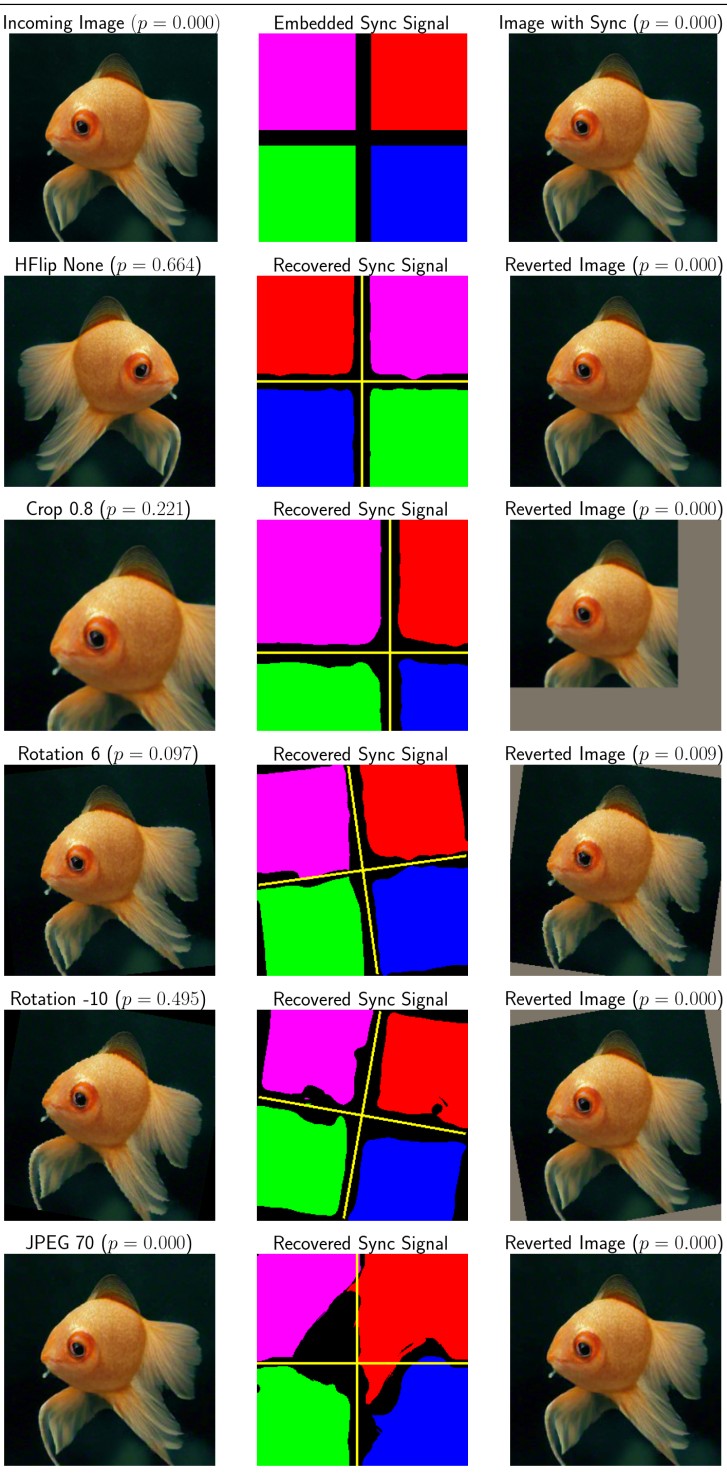

Figure 8: Visualization of our synchronization step (Sec. 3.2) on a real example from our experiments. In the four middle rows we see that the watermark detection would have failed on original geometrically transformed images, but has eventually succeeded after the synchronization signal was detected and reverted. In the bottom row we see that a valuemetric transformation can disrupt the signal—in this case a JPEG compression. While this did not hamper detection in this example, it can be problematic in practice as evidenced by the drop in valuemetric robustness with synchronization shown in Table 2.

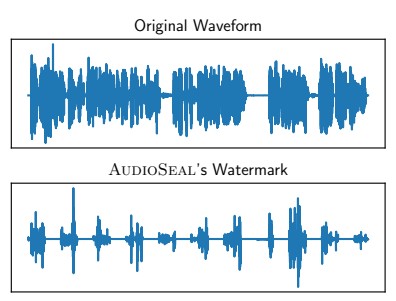

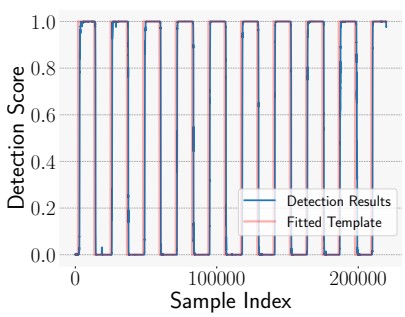

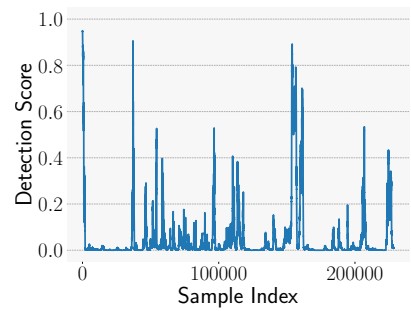

(a) Visualization of the synchronization mask pattern

(b) Example of successful detection after a small temporal crop

(c) Example of unsuccessful detection after $1.05\times$ speedup

Figure 9: The audio watermark synchronization method we attempted to incorporate. (a) how we embed the watermark periodically in the audio; (b) successful case with a clear periodic pattern detected when the audio is cropped for the first $0.84$ seconds; (c) a $1.05\times$ speedup creates a detection signal too noisy to reliably extract the synchronization pattern.

Given an incoming image, we first obtain and postprocess the predictions of $L$. Namely, for each pixel, we take the closest message in Hamming distance from the four fixed messages above, as long as the Hamming distance is below 6 bits and the probability of the pixel being watermarked as predicted by $L$ is above $0.5$. Then, as a heuristic, we proceed only if we found at least one pixel for each of the 4 messages, and if the total area of the pixels with messages is at least $70\%$ of the image—attempting to estimate the transformation otherwise proved too unreliable.

To estimate the transformation, we sweep over rotations in $[-20, 20]$ degrees, for each rotation *rotate the grid of extracted messages back* by the inverse rotation, and attempt to find the best-fitting pair $(i, j)$ such that the row $i$ of the rotated message grid best separates pixels with messages $(m_1, m_3)$ as well as $(m_2, m_4)$, and the column $j$ of the rotated message grid best separates pixels with messages $(m_1, m_2)$ as well as $(m_3, m_4)$. For example, to find $j$ that best separates $(m_1, m_2)$, we compute the *cost* of each candidate $j'$ as the number of *wrongly positioned* pixels, i.e., pixels with message $m_1$ that are to the right of $j'$ and pixels with message $m_2$ that are to the left of $j'$. We repeat the same cost computation for the horizontally flipped message grid: if we find that this leads to a lower cost, we estimate that the image was flipped. The lowest-cost estimate for $j$ and the estimate if the image was flipped or not are then aggregated over $(m_1, m_2)$ and $(m_3, m_4)$ proportionally to the number of pixels with these messages in the image. The process for $i$ is analogous, with the final result for the fixed suspect rotation being the tuple $(i, j, isFlipped, cost)$.

We finally take such tuple with the minimal *cost*, returning the corresponding rotation and $(i, j, isFlipped)$ as our final estimate that we later revert.

**Examples.** Complementing Fig. 4, in Fig. 8 we show real examples of recovered synchronization patterns and estimated transformations for horizontal flip, positive and negative rotation, crop, as well as Gaussian noise, that we found to be the most challenging valuemetric transformation in terms of disturbing the synchronization signal.

### D.2 Audio Synchronization with Localized Audio Watermarking

As explained in Sec. 5, we attempted to use the localization property of AudioSeal (San Roman et al., 2024) for synchronization similar to Sec. 3.2, but this approach proved less successful than with images. We describe below the method and experiments supporting this claim and summarize the results in Fig. 9.

**Synchronization approach and transformation detection.** We use AudioSeal's watermark embedder to embed watermarks with a periodic mask pattern across the audio signal. This enables detection of transformations like time-stretching and phase shifts through cross-correlation analysis. We applied a square wave template with periods of 6 frames, each frame being 1920 samples at 24kHz, as in MIMI.

When audio undergoes transformations, the periodic pattern distorts predictably. Through cross-correlation between the detection signal and template patterns, we can estimate the speedup factor (identified by the period maximizing cross-correlation) and phase shift (located by finding optimal alignment). Once estimated, we can invert the transformation by resampling to original speed and applying phase correction. More specifically, in our implementation, the detection results are first downsampled by a factor of 8 and we sweep the template period from $0.5\times$ to $1.5\times$ the nominal half-period

(6 frames = 0.48s) in coarse steps of 10 samples, then refine over $\pm 10$ samples around the best match; this two-stage cross-correlation yields precise estimates of speedup (from the best period) and phase shift, which are used to resample back to 24kHz and correct the alignment.

**Challenges and limitations.** As shown in Fig. 9(c), the detection signal extracted from AudioSeal is not yet robust enough for general use, as it fails to detect the watermark reliably after a speedup of $1.05\times$. Future work could explore more sophisticated synchronization methods.

# E Experimental Details

## E.1 Omitted Details of Image RCC Evaluation and Finetuning

Here we provide more details on our RCC evaluation experiments shown above in Table 1 and details related to RCC finetuning introduced in Sec. 3.1.

**RCC evaluation.** For Table 1, for simplicity, we re-use the watermarked BASE model of TAMING from our main experiments. We confirmed that running the non-watermarked version results in very similar values. We use the full set of valuemetric and geometric transformations as in the main experiments, and for each transformation use the same parameter that was chosen for summarized scores in Table 2, as detailed below.

**RCC finetuning.** To complete our RCC finetuning description from Sec. 3.1 we provide the omitted details. The finetuning is done for 10 epochs with distributed data parallel training on 16 V100 GPUs (TAMING, training takes 2h) and 8 H200 GPUs (CHAMELEON, training takes 2.5h). We use the Adam optimizer (Kingma & Ba, 2015) with a learning rate of $10^{-4}$, multiplied by a factor of 0.9 each epoch (StepLR). We use a total batch size across all GPUs of 64 (4 per gpu for TAMING and 8 per gpu for CHAMELEON), and always set $\lambda = 1$. As noted above, we use a set of augmentations $\mathcal{A}$ to improve robustness of our watermark to transformations and attacks. JPEG is not differentiable, therefore we backpropagate only through the difference between the uncompressed and compressed images (straight-through estimator): $x' = x_{\mathrm{aug}} + \mathrm{nograd}(x_{\mathrm{aug,JPEG}} - x_{\mathrm{aug}})$ (Yin et al., 2019; Zhang et al., 2021). We define three progressively harder sets: $\mathcal{A}_1, \mathcal{A}_2, \mathcal{A}_3$, and use no augmentations for 1 epoch, then $\mathcal{A}_1$ for 1 epochs, then $\mathcal{A}_2$ for 4 epochs, and finally $\mathcal{A}_3$ for the last 4 epochs.

$\mathcal{A}_1$ consists of JPEG compression with qualities $\{90, 80, 70\}$, Gaussian blur with kernel sizes $\{1, 3\}$, Gaussian noise with standard deviations $\{0.005, 0.01, 0.015, 0.02\}$, Brigthening with factors $\{1.0, 1.1, 1.2\}$, Rotation with angles $\{-1, 1\}$ degrees, and Cropping with percent kept from $\{80, 90\}$. $\mathcal{A}_2$ uses JPEG with qualities $\{80, 60, 40\}$, Gaussian blur with kernel sizes $\{3, 5\}$, Gaussian noise with standard deviations $\{0.02, 0.04, 0.06\}$, Brigthening with factors $\{1.2, 1.3, 1.4\}$, Rotation with angles $\{-3, -2, -1, 1, 2, 3\}$ degrees, and Cropping with percent kept from $\{50, 60, 70, 80, 90\}$. Finally, $\mathcal{A}_3$ uses JPEG with qualities $\{40, 30, 20\}$, Gaussian blur with kernel sizes $\{5, 7, 9\}$, Gaussian noise with standard deviations $\{0.06, 0.08, 0.1\}$, Brigthening with factors $\{1.4, 1.7, 2.0\}$, and the same geometric augmentations as in $\mathcal{A}_2$.

## E.2 Details of Main Experiments

We provide full details of our main experiments (Sec. 4.1 and Sec. 4.2), expanding on the information provided in the main paper.

**Models.** For TAMING, we use the VQGAN IMAGENET (F=16), 16384 version available in the authors' repository. For CHAMELEON, we use the 7B model. Since the open-weight version does not include image generation capabilities (as noted in the original paper), we obtained the necessary weights directly from the authors. Alternatively, image generation with CHAMELEON can be approximated using the Anole model and its associated repository: https://github.com/GAIR-NLP/anole, though we note that its output quality is somewhat lower.

**Parameters.** The results in Fig. 5, Fig. 6 and Table 2 are obtained from the same experiment, repeated on TAMING and CHAMELEON. For TAMING we set $\delta = 2, \gamma = 0.25, h = 1$ and evaluate (for each transformation/attack) on 1000 generations, 100 per each of the following ImageNet class indices: $[1, 9, 232, 340, 568, 656, 703, 814, 937, 975]$. For CHAMELEON we set $\delta = 2, \gamma = 0.25, h = 0$. We again use 1000 generations, conditioning the model on a text prompt each time. Following the standard protocol in the literature (Ramesh et al., 2021; 2022; Rombach et al., 2022; Saharia et al., 2022) we use the prompts from the validation set of MS-COCO (Lin et al., 2014). To do so, we first retrieve all the captions from the validation set, keep only the first one for each image, and select the first 1000 (or 5000 when computing FID for CHAMELEON). While we did not benchmark this in detail, the computational overhead of our watermark matches

that of the LLM watermarking scheme we inherit from *and* the localized watermark we use as the synchronization signal. As discussed in Sec. 6 more thoroughly integrating these two components could also make our watermark more efficient. A single run (e.g., BASE with all (augmentation, parameter) pairs detailed below on 1000 generations) with TAMING was executed on 25 V100 GPUs, lasting ≈30 minutes for BASE, FT, FT+AUGS and ≈1.5h for FT+AUGS+SYNC. For CHAMELEON, we use 10 H200 GPUs (50 for FT+AUGS+SYNC), taking similar time as for TAMING.

**Split stratification.** As noted in the literature (Esser et al., 2021; Yu et al., 2021), trained VQGANs often suffer from *low codebook utilization*, meaning that a certain percentage of the codebook is effectively not used and those codes (*dead codes*) are in practice never emitted by the transformer nor used when tokenizing images. While later work addresses this issue (Zhu et al., 2024), the VQGAN used in TAMING and in our experiments suffers from this issue and has only 971 *alive codes*, despite the codebook size of 16384.

This can affect the soundness of the watermark. Namely, the null hypothesis assumes that the ratio of green tokens in content produced without the use of the model $\mathcal{M}$ is $\gamma$. However, if the number of alive codes $n_{\text{alive}}$ is much smaller than the codebook size $|V|$, there is a non-negligible chance that choosing the set of green tokens as a uniformly random subset of $|V|$ of size $\gamma|V|$ results in significantly more or less than $\gamma n_{\text{alive}}$ *alive* green tokens. As these are the only tokens emitted by the tokenizer in practice, the green ratio under the null hypothesis can thus be significantly different from $\gamma$, making Eq. (2) inaccurate. While for $h > 0$ we can hope that this effect averages out across different contexts (as the expected alive green ratio is still $\gamma$), for $h = 0$ (fixed red/green split) this can introduce a constant bias.

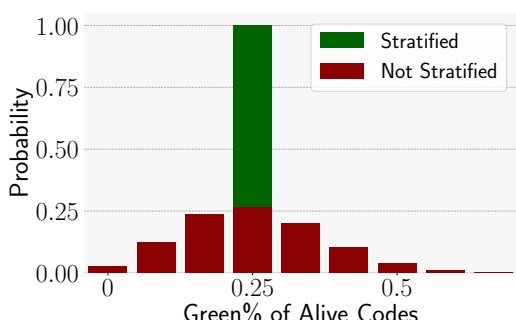

Figure 10: When the number of alive codes $n_{\text{alive}}$ is much smaller than the codebook size $|V|$, green/red splits may not correspond to the expected ratio of $\gamma$. In this figure, $n_{\text{alive}} = 12, |V| = 192, \gamma = 0.25$. Stratification, i.e., separate splitting of red and green tokens, resolves this issue.

In particular, assume for simplicity that $\gamma|V|$ and $\gamma n_{\text{alive}}$ are both integers. The probability $P_g$ that a uniformly random split of $|V|$ into $\gamma|V|$ green and $|V| - \gamma|V|$ red tokens results in *exactly g* green tokens among alive ones is not given by a binomial distribution but by a hypergeometric distribution:

$$P_g(|V|, n_{\text{alive}}, \gamma) = \frac{\binom{\gamma|V|}{g} \cdot \binom{|V|-\gamma|V|}{n_{\text{alive}}-g}}{\binom{|V|}{n_{\text{alive}}}}.$$ (10)

In Fig. 10 we plot the distribution of *actual green ratios* of green tokens, i.e., compute $P_g(|V|, n_{\text{alive}}, \gamma)/n_{\text{alive}}$ for different values of $g$, for $|V| = 192, n_{\text{alive}} = 12, \gamma = 0.25$ (corresponding roughly to the ratio $n_{\text{alive}}/|V|$ of the VQGAN we use in our experiments). We see (*in red*) that there is in fact only ≈25% chance that the green ratio among alive tokens is correctly set to $\gamma$. To resolve this, for TAMING we use a *stratified* split, i.e., we separately sample a red/green split on alive and dead codes, ensuring that the green ratio of alive tokens is exactly $\gamma$ (*green in Fig. 10*).

An interesting question, to the best of our knowledge not explored before, is if similar effects can be observed in LLM watermarking. For example, a uniformly random split of a large multilingual vocabulary may introduce a particularly biased split on e.g., Cyrillic tokens, which are effectively the majority of the alive ones when the LLM is prompted to write in a language that uses the Cyrillic script. Especially for $h = 0$, this may point at unfairness towards certain subdomains, where for a particular subdomain the watermark is overly conservative or more importantly has a much higher FPR than stated theoretically.

**Image transformations.** We next list all image transformations and their parameters used in our main experiments. We evaluate 90 variants (the original image and 89 transformations described below) for each image, i.e., 90,000 images in total per evaluation. For valuemetric transformations we use:

- Gaussian Blur: kernel sizes $[\underline{0}, 1, 3, 5, 7, \mathbf{9}, 11, 13, 15, 17, 19]$.
- Gaussian Noise: standard deviations $[\underline{0}, 0.025, 0.05, 0.075, \mathbf{0.1}, 0.125, 0.15, 0.175, 0.2]$.
- JPEG Compression: quality factors $[\underline{100}, 95, 85, 75, 65, 55, 45, 35, \mathbf{25}, 15, 5]$.
- Brighten: factors $[\underline{1}, 1.25, 1.5, 1.75, \mathbf{2}, 2.25, 2.5, 2.75, 3]$.

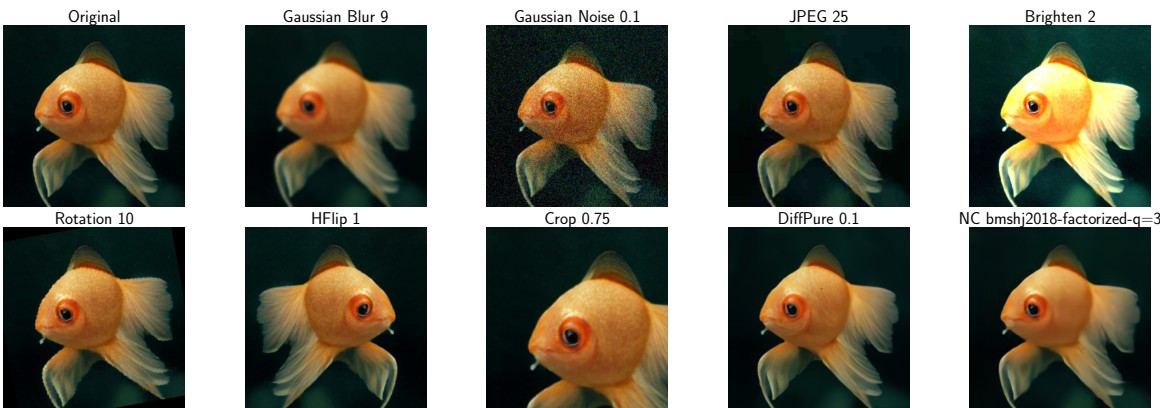

Figure 11: Examples of transformations with parameters used to compute the scores in Table 2.

For geometric transformations we use:

- Rotation: angles $[-20, -15, -10, -5, \underline{0}, 5, \mathbf{10}, 15, 20]$.

- Horizontal Flip: parameters $[\underline{0}, \mathbf{1}]$, where 1 indicates that a flip was performed.

- Crop: percent of the image kept $[\underline{1.0}, 0.95, 0.9, 0.85, 0.8, \mathbf{0.75}, 0.7, 0.65, 0.6, 0.55, 0.5]$, where we crop from the top-left corner of the image and then resize it back to the original size.

Finally, we use the following attacks:

- DiffPure: timesteps $[0.01, 0.05, \mathbf{0.1}, 0.2, 0.3]$ with the $256 \times 256$ ImageNet diffusion model used in the original attack of Nie et al. (2022).

- Neural Compression: a range of 22 models with different quality factors; see details below.

The underlined values above correspond to transformations that do not change the image (showing the maximum of robustness in each subplot of Fig. 6). The **bold** values are used to, following prior work (Wen et al., 2023), summarize the results to a single score per transformation/attack type in Table 2, where we average the 4 valuemetric scores and 3 geometric scores independently. For neural compression we describe how we compute the score below. Visual examples of each bold transformation/attack are shown on a real TAMING generation in Fig. 11.

**Neural compression.** For neural compression we use the following models from the `CompressAI` (Bégaint et al., 2020) library:

- BMSHJ18 (FACTORIZED) (Ballé et al., 2018) with quality factors $q \in \{1, 3, 6\}$.

- BMSHJ18 (HYPERPRIOR) (Ballé et al., 2018) with quality factors $q \in \{1, 3, 6\}$.

- CSTK20 (ANCHOR) (Cheng et al., 2020) with quality factors $q \in \{1, 3, 6\}$.

- CSTK20 (ATTENTION) (Cheng et al., 2020) with quality factors $q \in \{1, 3, 6\}$.

- MBT18 (Minnen et al., 2018) with quality factors $q \in \{1, 3, 6\}$.

- MBT18 (SCALE) (Minnen et al., 2018) with quality factors $q \in \{1, 3, 6\}$.

To sort these by compression strength we compute *bpp* (bits per pixel) as done in the library:

$$\frac{\sum_i \log L_i}{-\ln 2 \cdot n_{\text{pix}}}, \tag{11}$$

where $L$ is the likelihood vector and $n_{\text{pix}}$ is the number of pixels in the image. Empirically we observe bpp of around $0.1$ ($q = 1$), $0.3$ ($q = 3$), and $1.0$ ($q = 6$). When reporting a single score for neural compression we average the six scores with $q = 3$. Additionally, we evaluate the following four autoencoders from the `diffusers` (von Platen et al., 2022) library for which we compute bpp manually by considering the downscaling factor in the latent space, the latents size, and 16-bit/32-bit floating precision. We see that our calculations are consistent with the results of neural compressors from `CompressAI`:

- The Stable Diffusion VAE (Rombach et al., 2022) (`stabilityai/sd-vae-ft-ema`; SD VAE (FT-EMA)), with bpp 2.
- The Stable Diffusion XL VAE in half precision (Podell et al., 2024) (`madebyollin/sdxl-vae-fp16-fix`; SDXL VAE (FP16)), with bpp 1.
- The Deep Compression AE (Chen et al., 2024) (`mit-han-lab/dc-ae-f64c128-in-1.0-diffusers`; DC-AE), with bpp 1.
- The VAE of Flux (Labs, 2024) (from the `black-forest-labs/FLUX.1-schnell` pipeline; FLUX VAE), with bpp 1.

### E.3 Details of Audio Experiments

**Audio prompt generation.** We observed that MOSHI frequently generates brief responses and typically expects human interaction to continue the conversation. When using conventional text prompts such as those from Alpaca (Taori et al., 2023) or Open Assistant Conversations (Köpf et al., 2023) datasets, the model rarely produced audio outputs of sufficient length (e.g., 10 seconds). We therefore synthesized specialized prompts designed for this particular use case. These prompts are used when prompting the MOSHI model to generate (possibly watermarked) audio, such as in the experiments described in Sec. 5 and App. C and F.7.

To create a diverse collection of audio monologue topics, we leveraged LLAMA 3.1-8B-INSTRUCT to generate 1000 unique text prompts. We guided the model using a system+user template to produce concise single-sentence requests (each beginning with action verbs like "*Describe*", "*Talk about*", etc.) covering distinct subjects. We then filtered out near-duplicates by calculating pairwise Rouge-L scores (Lin, 2004) (using a threshold of 0.7) and eliminated texts that fell outside our desired length parameters. Representative examples include: "*Describe the life cycle of a butterfly and the symbolic meanings associated with it.*", "*Explain the process of photosynthesis in plants and its importance to ecosystems.*", or "*Discuss the cultural significance of traditional Japanese tea ceremonies.*". Finally, we converted these text prompts into audio using the SEAMLESSV2 (Barrault et al., 2023) (large) model, saving each resulting waveform alongside its corresponding source prompt. The resulting audio prompts average approximately 4 seconds in length.

**Audio transformations.** We evaluate robustness to a set of audio edits grouped into valuemetric, time-frequency, and neural compression transformations. When evaluating (e.g., in Table 3), each is applied with the following fixed strengths:

- Valuemetric:
  - Bandpass Filter: (300,3000), (500,5000), (1000,8000) Hz.
  - Highpass Filter: 100, 500, 1000 Hz.
  - Lowpass Filter: 1000, 3000, 8000 Hz.
  - Noise Injection (white): std = 0.001, 0.01, 0.05.
  - Pink Noise: std = 0.01, 0.05, 0.1.
  - Echo: (delay = 0.1 s, vol = 0.2), (0.3 s, 0.5), (0.5 s, 0.7).
  - Smooth: window fraction = 0.001, 0.005, 0.01.
  - Boost Audio: +50 %, +90 %.
  - Duck Audio: −50 %, −90 %.
  - MP3 Compression: bitrate = 16, 64, 128 kbps.
- Time-frequency:
  - Speed: factor = 0.75, 0.9, 1.0, 1.1, 1.25.
  - Temporal Crop: keep 50 %, 70 %, 90 % of duration.
  - Time Shift: shift = 10 ms, 20 ms, 40 ms.
  - Up/Down Resample: intermediate = 24 kHz, 36 kHz, 48 kHz.
- Neural Compression:
  - DAC Compression (24 kHz): full model pass.
  - EnCodec Compression (24 kHz): full model pass.

We use the same implementation as in AudioSeal (San Roman et al., 2024) when the augmentations are available. For DAC (Kumar et al., 2023) and EnCodec (Défossez et al., 2022) we use the official models at 24 kHz. We also provide examples of such augmentations in the supplementary material.

**RCC finetuning.** We perform fine-tuning for 200 epochs with 1000 steps per epoch on batches of 64 audio clips of 10-seconds from VoxPopuli (Wang et al., 2021), using 2 H200 GPUs for 1 day. We use the AdamW optimizer (Kingma & Ba, 2015) with a base learning rate of $2 \times 10^{-5}$, linear warmup over 5 epochs, and cosine annealing down to $2 \times 10^{-7}$. We set $\lambda$ to 0.01 for the regularization loss in the FT+AUGS model (with transformations), while using 0.001 in the FT model (without transformations). This regularization loss is the Multi-Resolution STFT loss between the audios reconstructed either with the original decoder $D_0$ or the finetuned decoder $D$. Following notations from Sec. 3.1, the RCC-loss is the MSE loss between $z$, the soft latents before the projection and quantization step, and $z'$, the soft latents generated by the encoder replica $E'$. To improve robustness for the FT+AUGS model, we apply augmentations $\mathcal{A}$ from the start, sampling one augmentation per batch. The augmentations are chosen randomly at each step, and the parameters are sampled uniformly from the ranges below:

- Lowpass filter: cutoff 2000-6000 Hz
- Highpass filter: cutoff 200-600 Hz
- White noise injection: std 0.001-0.01
- Pink noise: std 0.001-0.01
- Smooth: window fraction 0.001-0.005
- Time shift: 0.3-10 ms

## F  Additional Experimental Results

In this section, we present additional experimental results: ablations of RCC fine-tuning (App. F.1), investigations of different watermarking parameters (App. F.2), additional results for CHAMELEON on token match, watermark power, quality, and robustness (App. F.3), evaluation of decoder quality via PSNR comparisons (App. F.4), validation of statistical test correctness (App. F.5), comparison to generation-time watermarks (App. F.6), and omitted audio results (App. F.7).

### F.1  Finetuning Ablations

We train five more finetunes of TAMING to test the influence of different parameters:

- $\lambda = 10$ uses a higher regularization weight, i.e., puts less weight on the RCC loss.

- $\lambda = 0.1$ uses a lower regularization weight, i.e., puts more weight on the RCC loss.

- $lr = 10^{-5}$ uses a lower learning rate.

- $lr = 10^{-3}$ uses a higher learning rate.

- FT+AUGS_ALL finetunes all components of the VQGAN, including the codebook.

The results are presented in Table 5 and visual examples in Fig. 12 where the first row shows a detokenized output and the second row zooms in on the top-left $64 \times 64$ pixel region. Our baseline here is our FT+AUGS variant (we do not consider synchronization in this experiment). We see that increasing $\lambda$ to 10 slightly degrades the results (likely within the level of experimental noise) and reducing the learning rate reduces robustness. We also experimented with training for more epochs with a lower learning rate, but this did not lead to better results than our best variant.

Table 5: TPR at 1% FPR of finetuning ablations on TAMING in the setting of Table 2. The ablations are described in App. F.1. Three of the variants improve the results; however, Fig. 12 shows that they also lead to significant degradation in image quality.

|  | None | Valuemetric | Geometric | Adversarial Purification | Neural Compression |
|---|---|---|---|---|---|
| BASE | 0.99 | 0.26 | 0.01 | 0.43 | 0.48 |
| FT+AUGS | 1.00 | 0.92 | 0.01 | 0.70 | 0.79 |
| $\lambda = 10$ | 1.00 | 0.91 | 0.01 | 0.68 | 0.77 |
| $\lambda = 0.1$ | 1.00 | 0.98 | 0.01 | 0.85 | 0.96 |
| $lr = 10^{-5}$ | 0.99 | 0.75 | 0.01 | 0.63 | 0.76 |
| $lr = 10^{-3}$ | 0.98 | 0.98 | 0.01 | 0.81 | 0.78 |
| FT+AUGS_ALL | 1.00 | 1.00 | 0.02 | 0.85 | 0.98 |

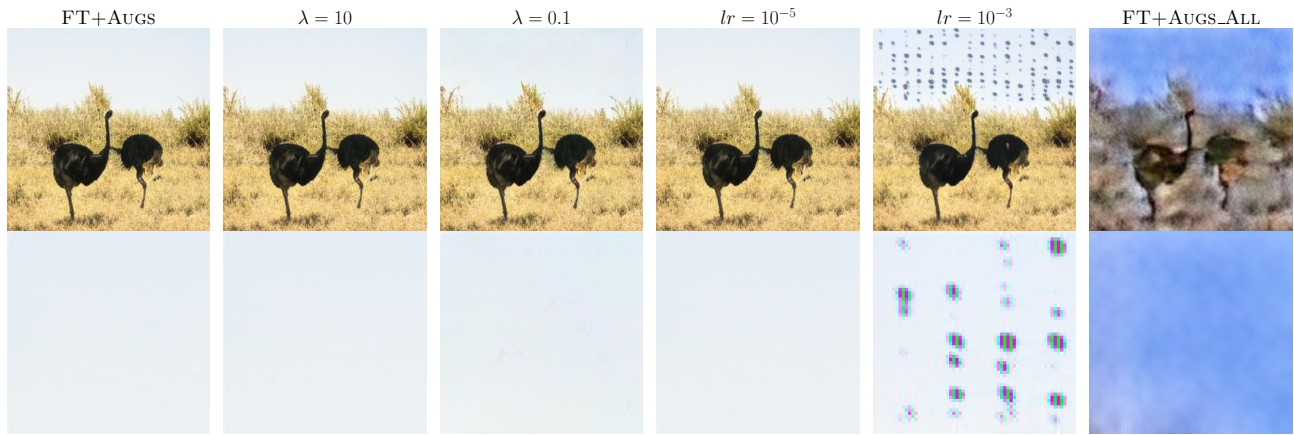

Figure 12: Visual examples of images produced by finetuning ablations introduced in App. F.1. The first row shows a detokenized image (based on the same token sequence as we fix the seed) and the second row zooms in on the top-left $64 \times 64$ pixel region to more clearly show artifacts. We see that the $\lambda = 0.1$, $\alpha = 10^{-3}$, and FT+AUGS_ALL variants lead to significant degradation in image quality.

We also see that $\lambda = 0.1$, $\alpha = 10^{-3}$, and FT+AUGS_ALL lead to much better results. However, the visual results in Fig. 12 show that all three of these variants lead to degradation in image quality. For FT+AUGS_ALL this is the most evident, which motivates our discussion above regarding the importance of carefully choosing which modules to finetune. The artifacts in $\alpha = 10^{-3}$ clearly show that the learning rate is too high for stable training. For $\lambda = 0.1$ the artifacts are clearly visible in the second row of the figure, showing that this setting puts too much weight on the RCC loss.

**Training on transformer-generated tokens.** Finally, we hypothesized that using token sequences generated by the transformer as the training set instead of tokenizations of ImageNet images would improve finetuning, as the former more closely matches the distribution of inputs that the detokenizer sees at evaluation time. Another experiment we tried was using a mixture of these two token sequences. We did not observe any benefits of this approach.

### F.2 Watermark Parameters

We summarize our takeaways from explorations of the main watermarking parameters: context size $h$, strength $\delta$, green ratio $\gamma$, the choice of watermark context, and the partitioning strategy.

For the context size $h$, we observed that $h > 1$ generally led to non-robust watermarks: we experimented with both $h = 2$ and $h = 3$ and both the standard choice of watermark context (preceding $h$ tokens) and the image-specific one (the $h$ tokens spatially close, e.g., for $h = 3$ the tokens above the current token, to the left of it, and top-left of it in the image). Even after RCC finetuning these variants had lower robustness than $h \in \{0, 1\}$ even on BASE. This is in line with the intuitive understanding of this parameter pointed out in prior work (Kirchenbauer et al., 2023; Zhao et al., 2023): large $h$ makes watermark removal easier, as changing any of the preceding $h$ tokens changes the red/green split at the following token. On the other hand, low $h$ makes the watermark less secure, i.e., easier to forge. As noted above, we hypothesize that $h = 0$ is a more viable choice for images, as reverse-engineering of the watermark rules (as successfully done for $h = 0$ in text (Jovanović et al., 2024; Zhang et al., 2024c)) is likely much more difficult due to the complex image tokenizer being hidden.

Increasing watermark strength $\delta$ led to either an increase in FID or visible artifacts, while reducing it sacrificed watermark power. We found that $\delta = 2$ is for both our models the strongest watermark that does not degrade quality. Our experiments with $\delta = 4$ increased FID by more than 2 points, while using $\delta = 1$ significantly reduced watermark power. We found that $\gamma = 0.25$ leads to slightly better tradeoffs than $\gamma = 0.5$.

Finally, we briefly experimented with a semantic partitioning strategy, related to similar attempts for LLMs (Hou et al., 2023). In particular, instead of choosing green tokens $\mathcal{G}$ uniformly at random from the vocabulary at each partitioning step, we use k-means clustering to partition the hard embeddings of each token in the vocabulary into $k = 100$ clusters and assign colors such that all tokens in the same cluster have the same color, while keeping the overall green ratio at $\gamma$. In theory, this should make the watermark more robust to modifications that do not change the semantics, as the resulting change in the token would hopefully not leave the k-means cluster and thus remain green. On the other hand, it might make it harder for

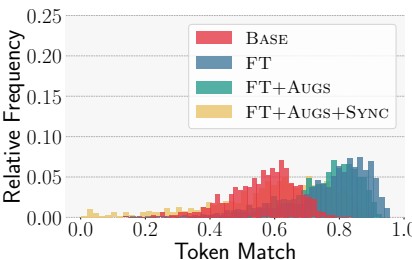
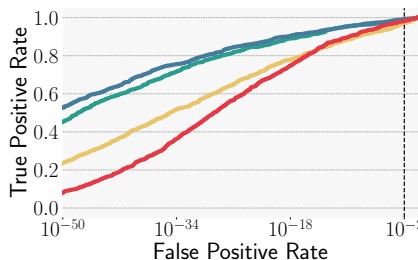

Figure 13: Token match and watermark power results for CHAMELEON. **Left**: Finetuning improves token match (Eq. (3)) between original and re-tokenized image tokens. **Right**: All variants achieve TPR ≈1 at FPR of 1%. Finetuning further boosts detection in low-FPR settings.

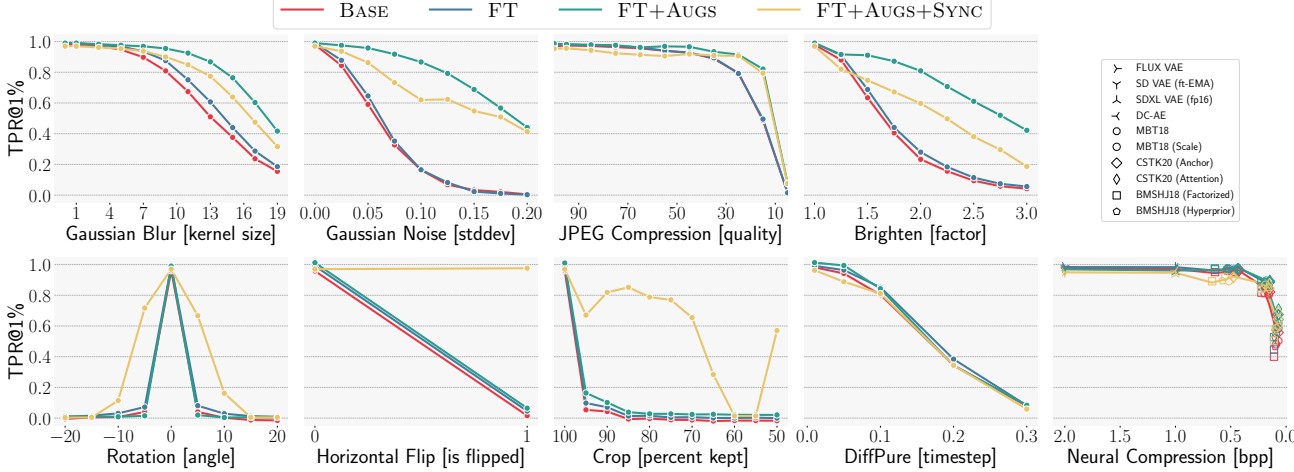

Figure 14: Robustness results for CHAMELEON, analogous to Fig. 6. **Top:** RCC finetuning improves robustness to valuemetric transformations. **Bottom Left:** Watermark synchronization unlocks robustness to geometric transformations. **Bottom Right:** Our watermark is also fairly robust to realistic strengths of diffusion purification and neural compression.

the decoding-time watermark to replace a green token with a red one, as all tokens from the same k-means cluster, which may be good alternatives, are also red. In practice, we consistently observed higher robustness but at the cost of significant quality degradation. More work is needed to understand how to find a good tradeoff with this approach.

### F.3    Extended Results for CHAMELEON

In Fig. 13 we present token match and watermark power results for CHAMELEON, complementing the TAMING results presented above in Fig. 5. Similarly, in Fig. 14 we present robustness results for CHAMELEON, similar to those in Fig. 6.

Additionally, we report FID results on CHAMELEON using 50,000 images as for TAMING, this time 10 independent generations per each of the 5000 COCO validation prompts (see App. E.2). As for TAMING we find that none of BASE, FT, or FT+AUGS exceed the unwatermarked FID of 19.7, which in this case also holds for FT+AUGS+SYNC[2].

### F.4    Evaluation of the Decoder's Quality

While FID assesses the quality of the generative model by comparing distributions of generated and real images, it does not allow direct comparison between individual images. In our case, however, we have access to both the original and fine-tuned decoders, enabling a more targeted evaluation of how fine-tuning affects the decoded outputs. To this end, we compute the average PSNR over 1000 image pairs as a direct measure of image-level similarity. In each pair, one image is decoded using the decoder from BASE, and the other using the decoder from FT or FT+AUGS, or using both the decoder

---

[2]For both models, we also visually confirm the quality of the generated images and the intensity of the diffs induced by the watermark—see App. G for examples.

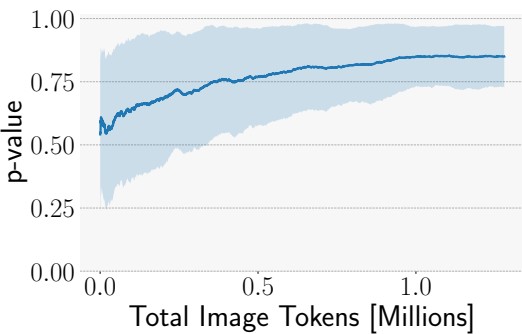 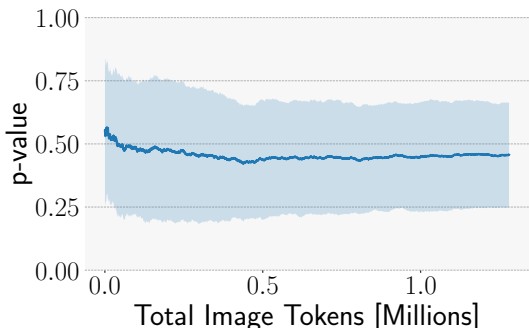

Figure 16: **Left**: On extremely long token sequences, the p-values on unwatermarked data increase as the real expected green ratio $\gamma' = 242/971$ is slightly below $\gamma = 0.25$ used in the test, making the test sound but overly conservative. **Right**: Using $\gamma'$ in the test rectifies this, and p-values behave as expected.

and synchronization step from FT+AUGS+SYNC. For TAMING, we obtain an average PSNR of 52.5 for FT, 49.7 for FT+AUGS, and 37.6 for FT+AUGS+SYNC. For CHAMELEON, the average PSNR is 56.1 for FT, 48.0 for FT+AUGS, and 39.5 for FT+AUGS+SYNC.

### F.5 Empirical Validation of Statistical Test Correctness

We empirically validate the correctness of our statistical test by computing p-values of our watermark detector on unwatermarked images. In Fig. 15 we show the distribution of such p-values on 50,000 unwatermarked images generated with the base model of TAMING, using the corresponding watermark ($\gamma = 0.25, h = 1$) across 10 different random seeds for the watermark. We observe that the distribution is roughly uniform; this holds also for each individual seed.

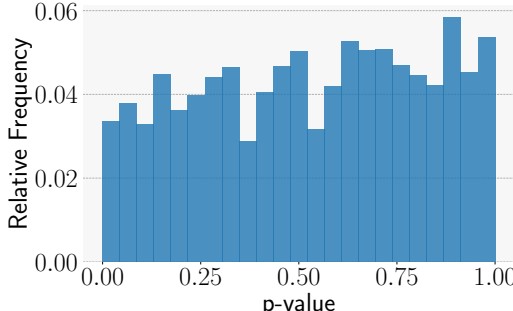

Figure 15: The distribution of p-values on unwatermarked images.

We push this investigation further by running our watermark detector on huge token sequences, as in Sander et al. (2024), despite those not being crucial to our usecase. Namely, for 10 random seeds, we 10 times independently concatenate 50,00 images to obtain a sequence of above 1M tokens, and compute the p-value on prefixes of this sequence of increasing length. We show the results in Fig. 16 (*left*, mean and standard deviation over 100 runs described above). Interestingly, while we would expect convergence around 0.5, the p-values of extremely long token sequences become as high as 0.8.

The reason for this is the relationship between the effective vocabulary size (i.e., the set of alive codes, see App. E) $n_{\text{alive}} = 971$ and the watermark parameter $\gamma = 0.25$. Specifically, as the number of green tokens $\gamma \cdot n_{\text{alive}} = 242.75$ in the vocabulary assumed by our test (Eq. (2)) is not an integer (which is never an issue for common values of $|V|$ and $\gamma$ in the literature), we chose to conservatively select 242 green tokens. This keeps the statistical test sound as the *real* expected green ratio $\gamma' = 242/971 \approx 0.2492$ is smaller than $\gamma = 0.25$, but may sacrifice some power. To further confirm that this is the main cause for the observed behavior, we repeat the experiment using $\gamma'$ instead of $\gamma$ in Eq. (2) and present the results in Fig. 16 (*right*). We observe that the p-values now converge to slightly below 0.5, which matches results in prior work (Sander et al., 2024).

### F.6 Comparison to Generation-time Watermarks

In Table 6 we present results on generation-time watermarks for image models. As no prior work studies autoregressive models, we show results for diffusion models. In particular, we study `stabilityai/stable-diffusion-2-1-base` (Rombach et al., 2022). For TREE-RING (Wen et al., 2023) we use the official implementation and set the watermark pattern to `ring`, `w_channels` to 3 and use 50 inference steps for generation and testing. For STABLE SIGNATURE (Fernandez et al., 2023b) we use the official implementation and set `unconditional_guidance_scale` to 9, `steps` to 50, and use `PLMSSampler` with `ddim_eta` set to 0. For GAUSSIAN SHADING (Yang et al., 2024) we use the official implementation, enable `chacha_encryption`, set the

Table 6: TPR at 1% FPR of different generation-time baselines. As these methods are not applicable to autoregressive image generation models, we show results on the `stabilityai/stable-diffusion-2-1-base` diffusion model. The transformations and attacks are the same as in Table 2

| | None | Valuemetric | Geometric | Adversarial Purification | Neural Compression |
|---|---|---|---|---|---|
| STABLE SIGNATURE (FERNANDEZ ET AL., 2023B) | 1.00 | 0.71 | 0.71 | 0.39 | 0.54 |
| TREE-RING (WEN ET AL., 2023) | 1.00 | 0.89 | 0.36 | 0.81 | 0.85 |
| GAUSSIAN SHADING (YANG ET AL., 2024) | 1.00 | 1.00 | 0.01 | 1.00 | 1.00 |

number of inference steps for generation and inversion to 50 and the number of bits to 256 with `channel_copy` set to 1 and `hw_copy` to 8.

We observe that each watermark is either fragile to geometric transformations or to adversarial purification and neural compression.

### F.7  Omitted Audio Results

**RCC finetuning.**  We presented in App. E.3 the details of our finetuning approach. Here we discuss the validation metrics and demonstrate how finetuning influences RCC and token match. We first evaluate the perceived audio quality of the generated samples using the PESQ and STOI metrics. After finetuning, we achieve a PESQ (Rix et al., 2001) score of 4.3 for both FT and FT+AUGS when compared to BASE samples. STOI (Taal et al., 2010) scores reach 0.98 for FT and 0.99 for FT+AUGS (we fixed the audio regularization loss weight to maintain approximately similar values for both approaches). The resulting audio is very hard to discriminate from the original, although we observe that it sometimes lead to light humming artifacts (see the audios in the supplementary material).

RCC and TM results are presented in Fig. 17. We observe significant token match improvements after finetuning when considering sequences of tokens generated by reconstructing 10-seconds VoxPopuli audios with the MIMI tokenizer. However, the finetuning process does not substantially improve token match for sequences generated by the MOSHI model with audio prompts. This may explain why finetuning does not significantly enhance watermark power for the MOSHI model, as observed in Table 3 of Sec. 5. A potential approach for improving watermark TPR would be to specifically finetune the model on generated sequences, which we leave for future work. (We attempted this approach for images but did not observe significant improvements, see App. F.1 for details.)

**Detailed robustness results.**  In Table 3 of the main paper, we report the average TPR at 1% FPR over multiple audio-specific augmentations. Table 7 provides the TPR for each individual augmentation that contributes to these averages, with a detailed view of how each transformation impacts watermark robustness. Table 7 also reports TPRs for different $\delta$ values. For instance, increasing $\delta$ to 4.0 pushes TPR above 0.9 across most augmentations, but the resulting audio quality deteriorates noticeably compared to lower-strength settings (see the audios in the supplementary material).

**Qualitative results.**  We provide in the supplementary material some audios corresponding to the prompts that were used for MOSHI, as well as MOSHI's completions when using BASE, FT, FT+AUGS, and for different values of $\delta$.

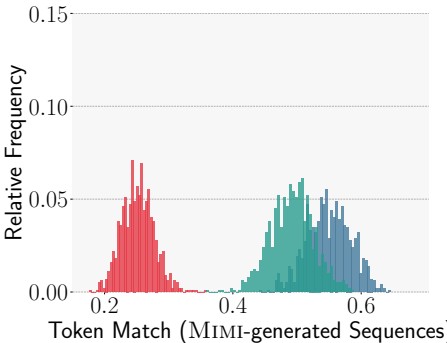 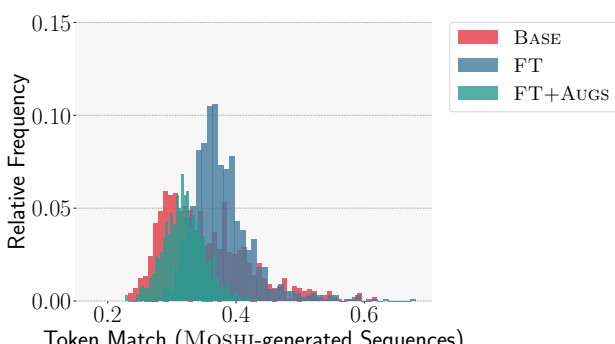

Figure 17: Token match histograms for sequences generated with MIMI (left) and MOSHI (right). We observe substantial consistency gains for MIMI-generated sequences and minimal change for MOSHI-generated ones.

Table 7: TPR at 1% FPR for the different tokenizer models, watermarking at different values for $\delta$ and different augmentation strengths. Audios are generated with MOSHI using the prompts described in App. E.3.

| Transformation | $\delta = 0.5$ | | | $\delta = 2.0$ | | | $\delta = 4.0$ | | |
| | BASE | FT | FT+AUGS | BASE | FT | FT+AUGS | BASE | FT | FT+AUGS |
|---|---|---|---|---|---|---|---|---|---|
| *Identity* | | | | | | | | | |
| Identity 0.0 | 0.31 | 0.57 | 0.46 | 0.98 | 0.99 | 0.99 | 0.99 | 0.99 | 0.99 |
| *Time-frequency* | | | | | | | | | |
| Speed 0.75 | 0.06 | 0.04 | 0.05 | 0.09 | 0.03 | 0.08 | 0.19 | 0.07 | 0.17 |
| Speed 0.9 | 0.07 | 0.07 | 0.08 | 0.27 | 0.20 | 0.30 | 0.63 | 0.49 | 0.70 |
| Speed 1.1 | 0.04 | 0.03 | 0.04 | 0.21 | 0.06 | 0.20 | 0.59 | 0.14 | 0.55 |
| Speed 1.25 | 0.02 | 0.01 | 0.02 | 0.09 | 0.02 | 0.07 | 0.26 | 0.04 | 0.19 |
| Crop 0.5 | 0.06 | 0.04 | 0.06 | 0.24 | 0.15 | 0.23 | 0.49 | 0.28 | 0.46 |
| Crop 0.7 | 0.06 | 0.05 | 0.07 | 0.30 | 0.19 | 0.33 | 0.59 | 0.35 | 0.54 |
| Crop 0.9 | 0.07 | 0.05 | 0.08 | 0.37 | 0.21 | 0.36 | 0.69 | 0.40 | 0.63 |
| Shift 10.0 | 0.09 | 0.09 | 0.14 | 0.56 | 0.33 | 0.80 | 0.91 | 0.68 | 0.97 |
| Shift 20.0 | 0.06 | 0.04 | 0.03 | 0.26 | 0.12 | 0.11 | 0.65 | 0.32 | 0.30 |
| Shift 40.0 | 0.06 | 0.08 | 0.03 | 0.17 | 0.24 | 0.12 | 0.48 | 0.64 | 0.39 |
| *Valuemetric* | | | | | | | | | |
| Bandpass (1000, 8000) | 0.03 | 0.04 | 0.16 | 0.10 | 0.15 | 0.45 | 0.28 | 0.38 | 0.85 |
| Bandpass (300, 3000) | 0.15 | 0.15 | 0.18 | 0.66 | 0.69 | 0.95 | 0.92 | 0.92 | 0.98 |
| Bandpass (500, 5000) | 0.10 | 0.17 | 0.18 | 0.45 | 0.53 | 0.94 | 0.83 | 0.85 | 0.98 |
| Boost 50 | 0.37 | 0.27 | 0.22 | 0.98 | 0.98 | 0.97 | 0.98 | 0.99 | 0.98 |
| Boost 90 | 0.21 | 0.19 | 0.14 | 0.96 | 0.93 | 0.88 | 0.98 | 0.98 | 0.98 |
| Duck 50 | 0.14 | 0.15 | 0.12 | 0.81 | 0.84 | 0.81 | 0.98 | 0.98 | 0.97 |
| Duck 90 | 0.11 | 0.09 | 0.14 | 0.48 | 0.30 | 0.58 | 0.83 | 0.63 | 0.90 |
| Echo (0.1, 0.2) | 0.12 | 0.20 | 0.13 | 0.87 | 0.97 | 0.93 | 0.98 | 0.99 | 0.98 |
| Echo (0.3, 0.5) | 0.05 | 0.09 | 0.07 | 0.55 | 0.79 | 0.60 | 0.89 | 0.97 | 0.93 |
| Echo (0.5, 0.7) | 0.04 | 0.05 | 0.04 | 0.38 | 0.56 | 0.40 | 0.77 | 0.91 | 0.79 |
| Highpass 100 | 0.27 | 0.49 | 0.40 | 0.98 | 0.99 | 0.99 | 0.98 | 0.99 | 0.99 |
| Highpass 1000 | 0.02 | 0.07 | 0.11 | 0.09 | 0.44 | 0.42 | 0.23 | 0.80 | 0.81 |
| Highpass 500 | 0.04 | 0.16 | 0.19 | 0.32 | 0.81 | 0.95 | 0.80 | 0.98 | 0.98 |
| Lowpass 1000 | 0.00 | 0.01 | 0.01 | 0.07 | 0.07 | 0.06 | 0.27 | 0.23 | 0.22 |
| Lowpass 3000 | 0.33 | 0.34 | 0.39 | 0.97 | 0.98 | 0.98 | 0.99 | 0.98 | 0.99 |
| Lowpass 8000 | 0.31 | 0.40 | 0.46 | 0.98 | 0.99 | 0.99 | 0.99 | 0.99 | 0.99 |
| MP3 128 | 0.28 | 0.48 | 0.44 | 0.97 | 0.99 | 0.98 | 0.99 | 0.99 | 0.99 |
| MP3 16 | 0.30 | 0.32 | 0.25 | 0.97 | 0.98 | 0.98 | 0.99 | 0.99 | 0.99 |
| MP3 64 | 0.27 | 0.43 | 0.43 | 0.97 | 0.99 | 0.99 | 0.98 | 0.99 | 0.99 |
| Noise 0.001 | 0.24 | 0.04 | 0.47 | 0.97 | 0.38 | 0.99 | 0.98 | 0.72 | 0.99 |
| Noise 0.01 | 0.03 | 0.01 | 0.31 | 0.50 | 0.01 | 0.98 | 0.86 | 0.01 | 0.98 |
| Noise 0.05 | 0.05 | 0.00 | 0.04 | 0.17 | 0.00 | 0.29 | 0.39 | 0.00 | 0.63 |
| Pink 0.01 | 0.24 | 0.49 | 0.43 | 0.97 | 0.98 | 0.99 | 0.98 | 0.98 | 0.99 |
| Pink 0.05 | 0.17 | 0.18 | 0.32 | 0.97 | 0.97 | 0.98 | 0.98 | 0.98 | 0.98 |
| Pink 0.1 | 0.07 | 0.06 | 0.14 | 0.93 | 0.80 | 0.96 | 0.97 | 0.96 | 0.97 |
| Smooth 0.001 | 0.06 | 0.04 | 0.10 | 0.44 | 0.24 | 0.79 | 0.83 | 0.52 | 0.96 |
| Smooth 0.005 | 0.01 | 0.00 | 0.10 | 0.02 | 0.01 | 0.58 | 0.07 | 0.01 | 0.90 |
| Smooth 0.01 | 0.01 | 0.01 | 0.07 | 0.02 | 0.02 | 0.38 | 0.04 | 0.03 | 0.76 |
| UpDown Res. 24000 | 0.31 | 0.57 | 0.46 | 0.98 | 0.99 | 0.99 | 0.99 | 0.99 | 0.99 |
| UpDown Res. 36000 | 0.30 | 0.52 | 0.46 | 0.98 | 0.99 | 0.99 | 0.99 | 1.00 | 0.99 |
| UpDown Res. 48000 | 0.31 | 0.52 | 0.46 | 0.98 | 0.99 | 0.99 | 0.99 | 1.00 | 0.99 |
| *Neural Compression* | | | | | | | | | |
| DAC 0.0 | 0.24 | 0.17 | 0.38 | 0.97 | 0.96 | 0.99 | 0.98 | 0.98 | 0.99 |
| EnCodec 0.0 | 0.07 | 0.06 | 0.09 | 0.62 | 0.55 | 0.69 | 0.91 | 0.91 | 0.94 |

# G   Qualitative Examples

In Fig. 18 and Fig. 19 we show qualitative examples of our watermark on images generated by TAMING (resp. CHAMELEON) and the post-hoc baselines previously evaluated in Sec. 4.2.

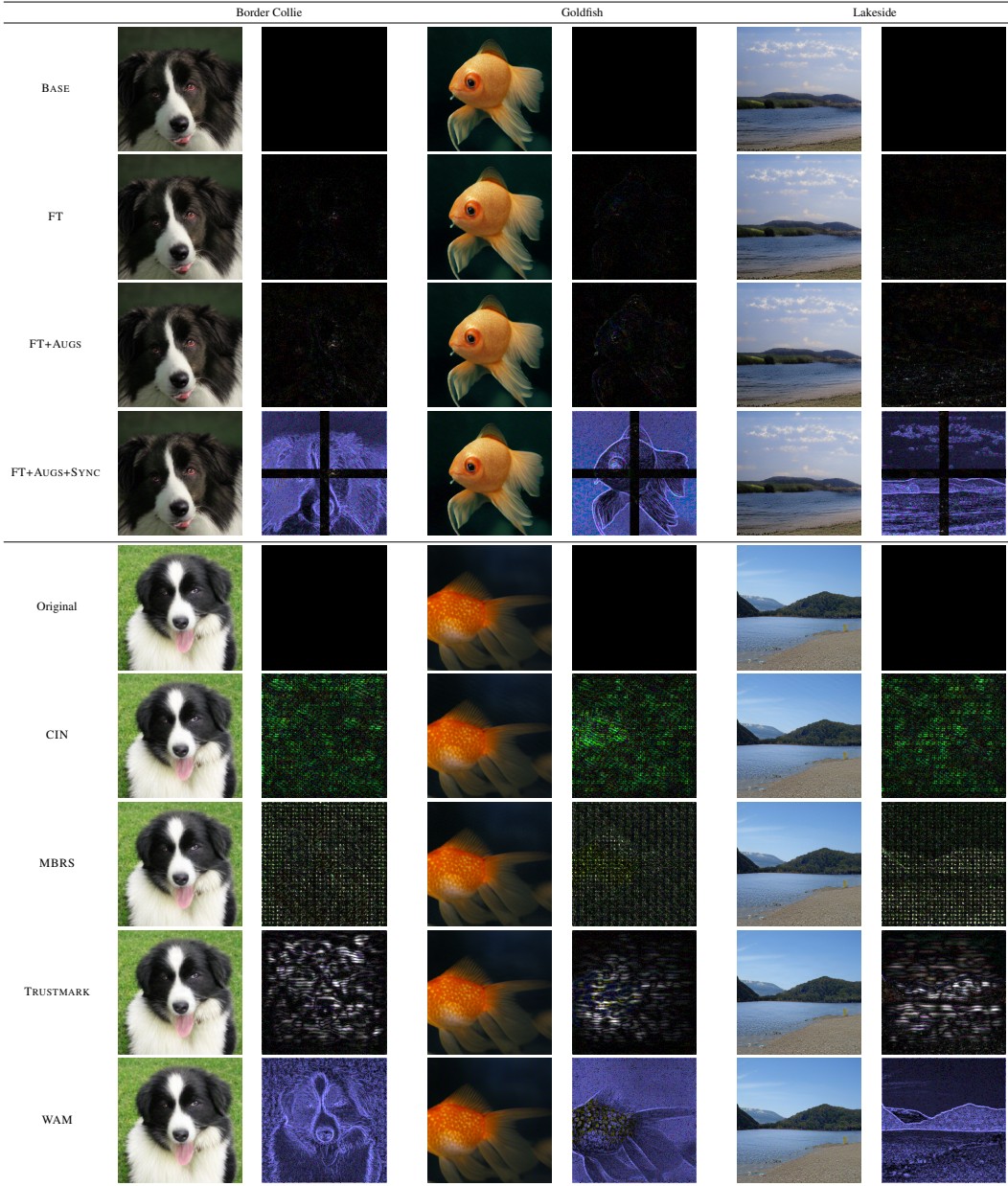

Figure 18: Qualitative results on TAMING with samples from 3 of the ImageNet-1k classes. The left column shows the images and the right column the diffs. For variants of our watermark (*top*) the diff is computed w.r.t. the BASE decoder (as there is no notion of an original unwatermarked image). For post-hoc baselines (*bottom*) the diff is computed w.r.t. the original image. All diffs are displayed with the same postprocessing applied for visibility, namely $\text{clip}(|a - b| \cdot 30, 0, 255)$, where $a$ and $b$ are pixel values of the two images in $[0, 255]$.

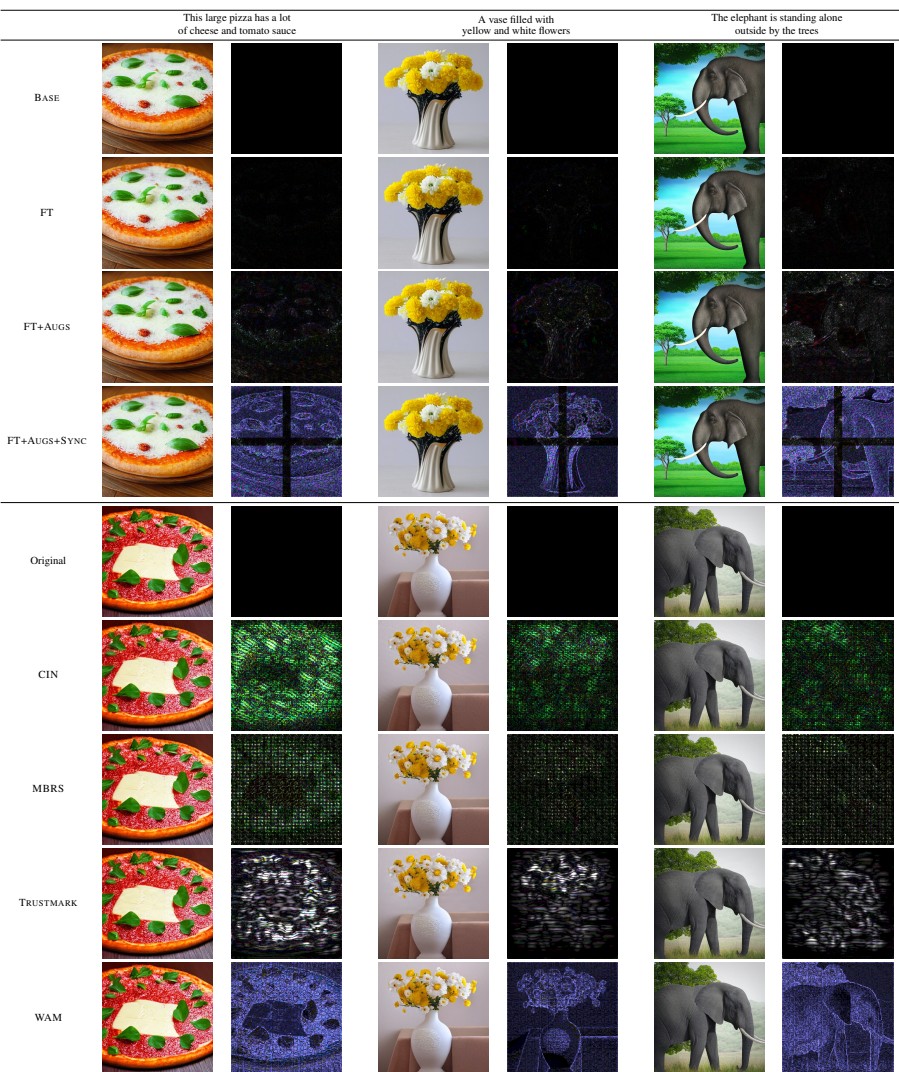

Figure 19: Qualitative results on CHAMELEON with 3 COCO validation prompts. The left column shows the images and the right column the diffs. For variants of our watermark (*top*) the diff is computed w.r.t. the BASE decoder (as there is no notion of an original unwatermarked image). For post-hoc baselines (*bottom*) the diff is computed w.r.t. the original image. All diffs are displayed with the same postprocessing applied for visibility, namely $\text{clip}(|a - b| \cdot 30, 0, 255)$, where $a$ and $b$ are pixel values of the two images in $[0, 255]$.

## H   On Joint Watermarking of Interleaved Modalities

In this section, we extend our discussion on joint watermarking of interleaved modalities from Sec. 4.3, provide omitted experimental details, and present extended experimental results.

**Experimental details.**   We query CHAMELEON with 1000 prompts, each generated from one of ImageNet-1K classes, asking the model to teach the user about the notion represented by the class label and illustrate it. We use $\gamma = 0.25$ and set $h = 0$ for images and $h = 1$ for text. Two example interactions are shown in App. H.1. For text corruption we explore percentages in $[0, 60]$. For Gaussian noise corruption of images, we use $\sigma \in \{0, 0.05, 0.1, 0.15, 0.2, 0.25, 0.3\}$.

**Full experimental results.**   Extended results are shown in Fig. 20. We observe that for high-quality images where the watermark signal is preserved, joint detection is always beneficial, while for highly corrupted images, it almost never is. Between these two, joint detection becomes beneficial starting from some text corruption level. The intuitive understanding is that joint detection that integrates a *better quality* signal is always beneficial. Perhaps unexpectedly, it can be also beneficial when signal is of slightly lower quality if it sufficiently increases the number of tokens. This is important as a single image consists of a large number of tokens, in particular 1024 for CHAMELEON, while the average text length in our experiment is $\approx 227$.

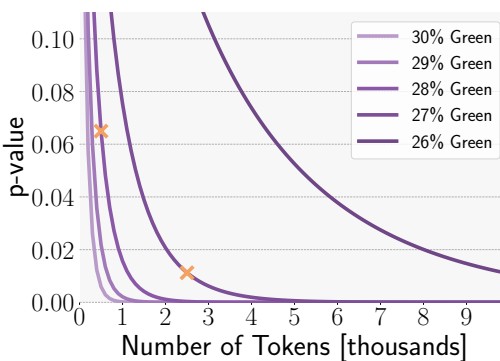
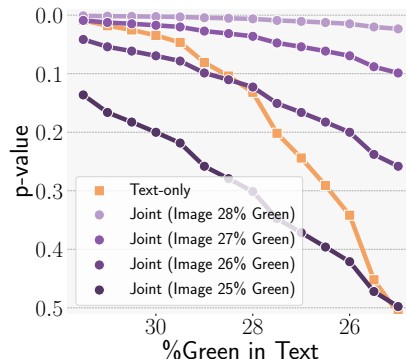

Figure 21: **Left**: Using more tokens at the same green token ratio reduces the p-value. As two orange crosses show, using $5\times$ more tokens ($500 \to 2500$) can improve the p-value even if the green ratio drops ($28\% \to 27\%$). **Right**: Simulated results similar to those in Fig. 7. Both axes are flipped to match Fig. 7: moving on the x-axis reduces the % of green tokens in text (*weaker signal*) and moving on the y-axis reduces the p-value (*stronger detection*). As images carry many tokens, merging e.g., text with $28\%$ green tokens with an image with $26\%$ green tokens still improves detection.

**When is joint detection beneficial?** First, to elaborate on our claim from the main paper that *scoring more equally watermarked tokens improves power*, we plot the p-value of the watermark detection as a function of token length for a *fixed green ratio* $S/(T - h)$ (Fig. 21, left). For each line, increasing the number of tokens can rapidly improve the detection p-value. To illustrate the point made above that increasing the number of tokens can be beneficial even if the green ratio drops, note the two orange crosses. Increasing the number of tokens fivefold ($500 \to 2500$) while reducing the green ratio from $28\%$ to $27\%$ has still dropped the p-value from around $0.06$ to almost $0.01$ which for FPR of $1\%$ is the passing threshold.

Rounding the average text length in our experiment to 256, the fivefold increase is exactly the average case that we observed ($256 \to 1024 + 256$). We simulate this case (a join of 256 text and 1024 image tokens) while fixing the green ratio for both modalities. Our results in Fig. 21 (right) show a similar trend as in our above

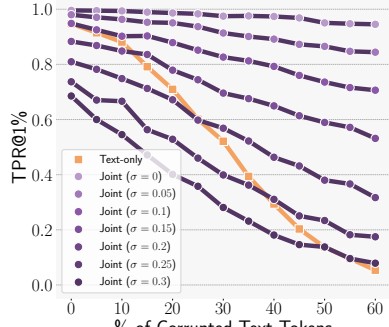

Figure 20: Full results on joint watermark detection, extending the results in Fig. 7.

experiment on real data. For example, merging text with $28\%$ green tokens with an image with only $26\%$ green tokens still improves the p-value.

These results illustrate the benefits of watermarking modalities jointly. In contrast, if text and images were watermarked separately, to reject the null hypothesis of *this article was not generated by my model $\mathcal{M}$* in a statistically sound way, we would need to take the maximum of the p-values of the two individual tests, i.e., the improvements seen above would not be possible.

**Outlook.** We hope this preliminary investigation opens the door to more detailed studies of cross-modal watermarking in future work. Other interesting questions include the comparison of the impact of different transformations/attacks on the watermark *across modalities*, or the relationship between using more tokens from the same modality (when possible)and from a new modality. Intuitively, as modality-specific vocabularies are disjoint, for already long sequences the former (same modality) may have a higher chance of introducing many duplicate tokens, which do not affect the p-value. On the other hand, the probability of duplicates within a sample may be higher for some modalities: in the first example in App. H.1, the image mostly consists of duplicate white tokens.

### H.1 Example Interactions

We show two interactions with CHAMELEON from our joint watermark experiment (Sec. 4.3), along with the corresponding watermark detection results. In the detector view, for both modalities, ■ denotes tokens scored green, ■ denotes tokens scored red, and □ those ignored due to duplication (e.g., white visual tokens in the first example, we use gray for visibility).

---

### 🧑 User Prompt

I am wondering about the following: **acoustic guitar**. Can you teach me something about it in a way that is easy to understand and also generate a picture of it?

---

### 🦎 Chameleon

Certainly, I'd be happy to help!

The acoustic guitar, as shown in the image below, is a popular instrument for playing music. It is made of a wooden body, a neck, and strings that are plucked or strummed to produce sound. The sound of an acoustic guitar is created by the vibrations of the strings, which are then amplified by the body of the guitar.

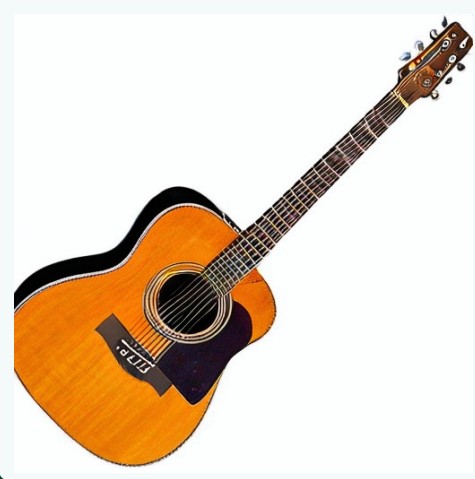

---

### 💧 Watermark Detector ($p = 1.7 \cdot 10^{-16}$)

Certainly, I'd be happy to help!

The acoustic guitar, as shown in the image below, is a popular instrument for playing music. It is made of a wooden body, a neck, and strings that are plucked or strummed to produce sound. The sound of an acoustic guitar is created by the vibrations of the strings, which are then amplified by the body of the guitar.

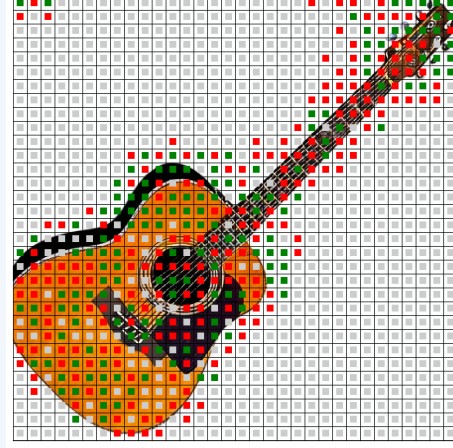

### 👤 User Prompt

I am wondering about the following: **thatch**. Can you teach me something about it in a way that is easy to understand and also generate a picture of it?

---

### 🦎 Chameleon

Sure! Thatch refers to the plant material used to cover roofs and walls in traditional or indigenous architecture. It is made from various types of grasses, leaves, or other vegetation. Here is an image of an example of thatch, used as roofing material in the construction of this house .Thatch is often made from locally available resources, such as reeds, rushes, or grasses, and is often woven or tied together to create a durable and effective roofing material. It requires regular maintenance to keep it in good condition and can last for many years with proper care. In some areas, thatch is still used today as a traditional building material, while in others, it has been largely replaced by modern roofing systems.

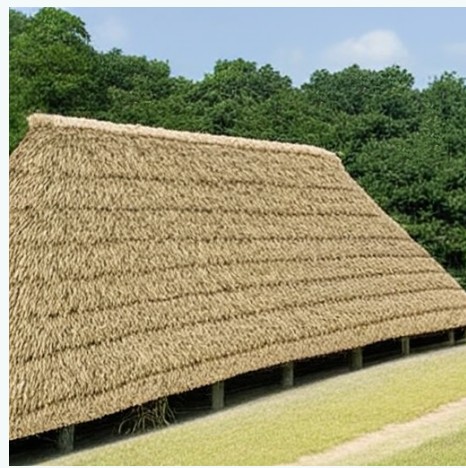

---

### 💧 Watermark Detector ($p = 1.2 \cdot 10^{-34}$)

Sure! Thatch refers to the plant material used to cover roofs and walls in traditional or indigenous architecture. It is made from various types of grasses, leaves, or other vegetation. Here is an image of an example of thatch, used as roofing material in the construction of this house .Thatch is often made from locally available resources, such as reeds, rushes, or grasses, and is often woven or tied together to create a durable and effective roofing material. It requires regular maintenance to keep it in good condition and can last for many years with proper care. In some areas, thatch is still used today as a traditional building material, while in others, it has been largely replaced by modern roofing systems.

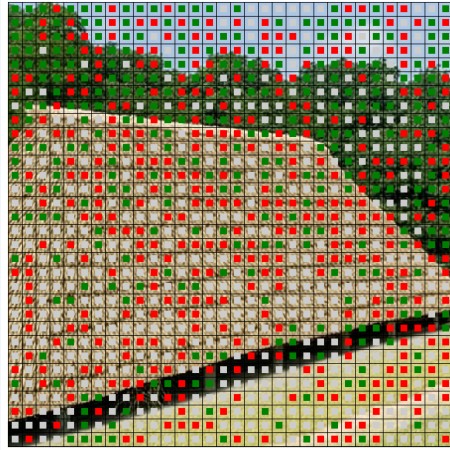

