# OpenReview forum: "Watermarking Autoregressive Image Generation"
_ICML.cc/2025/Workshop/TokShop — TokShop_

### Official Review · Reviewer_NLes · 2025-06-05

**Rating:** 8
**Confidence:** 2

**Review:**

This paper applies methods used for watermarking LLMs to autoregressive image generation. They identify a key challenge in this process: this method lacks reverse cycle-consistency, i.e. detokenizing a watermaked image alters the sequence and erases the watermark. They propose a way to mitigate it and do a large-scale evaluation of their model, showing good results and robustness.

I believe that the paper's contribution is a valuable one. This is not my area, but they claim to be the first to watermark autoregressive image generation and I have no reason to doubt that. It seems like a useful contribution in itself. The experimental framework and results all seem sound to me from what I can judge. I do think, however, that the paper could gain in clarity, even to an audience that may be more familiar with the topic. The paper could explain some important concepts in greater detail. Most strikingly, the method they rely on is explained as 'Generation-time methods directly alter generations to embed statistically detectable patterns'. This is a very short and imprecise explanation for a method that is so central to the paper. It is partially explained in the figure but that is also not easily resolvable. Describing this method, referring to the figure in detail, would be highly beneficial I believe. In addition, the paper is heavy in mathematical notations and abbreviations making it difficult to follow. For example, on line 174, h=1 is not intuitive to me and I have to go back to the definition of h on line 127 in the previous page. This happens a lot. Some notations are introduced but never referenced again in the main paper (e.g. ξ), or, not a notation, but the provider "Bob". I think a lot could be done to save efforts to the reader. Space can be made e.g. by avoiding the citation bomb drops of papers that are not really relevant, eg. on l086 col 2, l092 col 2 l148 col 1, etc.

Other things I find unclear (non-exhaustive list):
* Alice’s goal is out-of-model, generation-time, zero-bit watermarking (see Sec. 2) - unclear reference, first and only mention of the term zero-bit
* The LPIPS abbreviation is never introduced
* l429: "scoring more equally watermarked tokens"
* l 402, col 2: 'As all suspect text is used'

---

### Official Review · Reviewer_NohT · 2025-06-09
**Thorough study on a timely topic relevant to the workshop (token-level watermarking in autoregressive image generation)**

**Rating:** 8
**Confidence:** 4

**Review:**

**Summary:**

This paper introduces an approach for token-level watermarking of autoregressive image generation models, adapting LLM watermarking techniques (specifically KGW) to image tokens. The main contribution is a method that enables robust watermark detection even after common image transformations or specific adversarial attacks.

**Strengths:**

- Studies a timely topic and represents a novel extension of LLM watermarking to non-text modalities
- Proposes a finetuning procedure for image tokenization that improves RCC and significantly increases watermark power and robustness.
- Introduces a post-hoc watermark synchronization step, which achieves geometric robustness by leveraging off-the-shelf localized watermarking.


**Weaknesses:**

- The approach is not robust to composite attacks (e.g., transformation + purification), which the authors acknowledge and is a limitation shared by prior works
- The synchronization step involves heuristic search and message detection but lacks analysis of computational efficiency or failure cases under incorrect transformations.

**Comments:**
- In the approach, the watermark strength and context window settings are fixed; further analysis on their influence on detectability vs. quality trade-offs would be valuable (but I don't think it's necessarily a weakness).
- Given that synchronization involves multiple image transformations, it would help to know the runtime overhead in practical settings.

Since the submission is non-archival, I didn’t carefully go through the long appendix and other details, but the paper looks sound overall. I think it is worth disseminating at the workshop to get more people to think/discuss about watermarking and how it relates to tokenization.

---

### Decision · Program_Chairs · 2025-06-10

Accept